# Recovering the Lowest Layer of Deep Networks with High Threshold Activations

## Abstract

Giving provable guarantees for learning neural networks is a core challenge of machine learning theory. Most prior work gives parameter recovery guarantees for one hidden layer networks, however, the networks used in practice have multiple non-linear layers. In this work, we show how we can strengthen such results to deeper networks – we address the problem of uncovering the lowest layer in a deep neural network under the assumption that the lowest layer uses a high threshold before applying the activation, the upper network can be modeled as a well-behaved polynomial and the input distribution is gaussian.

## 1 Introduction

Understanding the landscape of learning neural networks has been a major challege in machine learning. Various works gives parameter recovery guarantees for simple one-hidden-layer networks where the hidden layer applies a non-linear activation $u$ after transforming the input $\boldsymbol{x}$ by a matrix $\mathbf{W}$, and the upper layer is the weighted sum operator: thus $f(\boldsymbol{x}) = \sum a_i u(\boldsymbol{w}_i^T \boldsymbol{x})$. However, the networks used in practice have multiple non-linear layers and it is not clear how to extend these known techniques to deeper networks.

We consider a multilayer neural network with the first layer activation $u$ and the layers above represented by an unknown polynomial $P$ such that it has non-zero non-linear components. More precisely, the function $f$ computed by the neural network is as follows:

$$f_{\mathbf{W}}(\boldsymbol{x}) = P(u(\boldsymbol{w}_1^T \boldsymbol{x}), u(\boldsymbol{w}_2^T \boldsymbol{x}), \ldots, u(\boldsymbol{w}_d^T \boldsymbol{x})) \text{ for } P(X_1, \ldots, X_d) = \sum_{\boldsymbol{r} \in \mathbb{Z}_+^d} c_{\boldsymbol{r}} \cdot \prod_j X_j^{r_j}.$$

We assume that the input $x$ is generated from the standard Gaussian distribution and there is an underlying true network (parameterized by some unknown $\mathbf{W}^*$)[1] from which the labels are generated.

In this work we strengthen previous results for one hidden layer networks to a larger class of functions representing the transform made by the upper layer functions if the lowest layer uses a high threshold (high bias term) before applying the activation: $u(a - t)$ instead of $u(a)$. Intuitively, a high threshold is looking for a high correlation of the input $a$ with a direction $\boldsymbol{w}_i^*$. Thus even if the function $f$ is applying a complex transform after the first layer, the identity of these high threshold directions may be preserved in the training data generated using $f$.

**Learning with linear terms in $P$.** Suppose $P$ has a linear component then we show that increasing the threshold $t$ in the lowest layer is equivalent to amplifying the coefficients of the linear part. Instead of dealing with the polynomial $P$ it turns out that we can roughly think of it as $P(\mu X_1, ..., \mu X_d)$ where $\mu$ decreases exponentially in $t$ ($\mu \approx e^{-t^2}$). As $\mu$ decreases it has the effect of diminishing the non-linear terms more strongly so that relatively the linear terms stand out. Taking advantage of this effect we manage to show that if $t$ exceeds a certain threshold the non linear terms drop in value enough so that the directions $\boldsymbol{w}_i$ can be learned by relatively simple methods. We show that we can get close to the $\boldsymbol{w}_i$ applying a simple variant of PCA. While an application of PCA can be thought of as finding principal directions as the local maxima of $\max_{||\boldsymbol{z}||=1} \mathbb{E}[f(\boldsymbol{x})(\boldsymbol{z}^T \boldsymbol{x})^2]$,

---

[1] We suppress $\mathbf{W}$ when it is clear from context.

we instead perform $\max_{\mathbb{E}[f(\boldsymbol{x})H_2(\boldsymbol{z}^T\boldsymbol{x})^2]=1} \mathbb{E}[f(\boldsymbol{x})H_4(\boldsymbol{z}^T\boldsymbol{x})^4]]^2$. If $\mathbf{W}^*$ has a constant condition number then the local maxima can be used to recover directions that are transforms of $\boldsymbol{w}_i$.

**Theorem 1** (informal version of Claim 2, Theorem 11). *If $t > c\sqrt{\log d}$ for large enough constant $c > 0$ and $P$ has linear terms with absolute value of coefficients at least $1/\mathsf{poly}(d)$ and all coefficients at most $O(1)$, we can recover the weight vector $\boldsymbol{w}_i$ within error $1/\mathsf{poly}(d)$ in time $\mathsf{poly}(d)$.*

These approximations of $\boldsymbol{w}_i$ obtained collectively can be further refined by looking at directions along which there is a high gradient in $f$; for monotone functions we show how in this way we can recover $\boldsymbol{w}_i$ exactly (or within any desired precision.

**Theorem 2.** *(informal version of Theorem 5) Under the conditions of the previous theorem, for monotone $P$, there exists a procedure to refine the angle to precision $\epsilon$ in time $\mathsf{poly}(1/\epsilon, d)$ starting from an estimate that is $1/\mathsf{poly}(d)$ close.*

The above mentioned theorems hold for $u$ being sign and ReLU.[3]

When $P$ is monotone and $u$ is the sign function, learning $W$ is equivalent to learning a union of half spaces. We learn $\mathbf{W}^*$ by learning sign of $P$ which is exactly the union of halfspaces $\boldsymbol{w}_i^T\boldsymbol{x} = t$. Thus our algorithm can also be viewed as a polynomial time algorithm for learning a union of large number of half spaces that are far from the origin – to our knowledge this is the first polynomial time algorithm for this problem but with this extra requirement (see earlier work Vempala (2010) for an exponential time algorithm). Refer to Appendix B.6 for more details.

Such linear components in $P$ may easily be present: consider for example the case where $P(X) = u(\boldsymbol{v}^T X - b)$ where $u$ is say the sigmoid or the logloss function. The taylor series of such functions has a linear component – note that since the linear term in the taylor expansion of $u(x)$ has coefficient $u'(0)$, for expansion of $u(x-b)$ it will be $u'(-b)$ which is $\Theta(e^{-b})$ in the case of sigmoid. In fact one may even have a tower (deep network) or such sigmoid/logloss layers and the linear components will still be present – unless they are made to cancel out precisely; however, the coefficients will drop exponentially in the depth of the networks and the threshold $b$.

**Sample complexity with low thresholds and no explicit linear terms.** Even if the threshold is not large or $P$ is not monotone, we show that $\mathbf{W}^*$ can be learned with a polynomial sample complexity (although possibly exponential time complexity) by finding directions that maximize the gradient of $f$.

**Theorem 3** (informal version of Corollary 1). *If $u$ is the sign function and $\boldsymbol{w}_i$'s are orthogonal then in $\mathsf{poly}(1/\epsilon, d)$ samples one can determine $\mathbf{W}^*$ within precision $\epsilon$ if the coefficient of the linear terms in $P(\mu(X_1 + 1), \mu(X_2 + 1), \mu(X_3 + 1), \ldots)$ is least $1/\mathsf{poly}(d)$*

**Learning without explicit linear terms.** We further provide evidence that $P$ may not even need to have the linear terms – under some restricted cases (section 4), we show how such linear terms may implicitly arise even though they may be entirely apparently absent. For instance consider the case when $P = \sum X_i X_j$ that does not have any linear terms. Under certain additional assumptions we show that one can recover $\boldsymbol{w}_i$ as long as the polynomial $P(\mu(X_1 + 1), \mu(X_2 + 1), \mu(X_3 + 1), ..)$ (where $\mu$ is $e^{-t}$ has linear terms components larger than the coefficients of the other terms). Note that this transform when applied to $P$ automatically introduces linear terms. Note that as the threshold increases applying this transform on $P$ has the effect of gathering linear components from all the different monomials in $P$ and penalizing the higher degree monomials. We show that if $\mathbf{W}^*$ is a sparse binary matrix then we can recover $\mathbf{W}^*$ when activation $u(a) = e^{\rho a}$ under certain assumptions about the structure of $P$. When we assume the coefficients are positive then these results extend for binary low $l_1$- norm vectors without any threshold. Lastly, we show that for even activations ($\forall a, u(a) = u(-a)$) under orthogonal weights, we can recover the weights with no threshold.

**Learning with high thresholds at deeper layers.** We also point out how such high threshold layers could potentially facilitate learning at any depth, not just at the lowest layer. If there is any cut in the network that takes inputs $X_1, \ldots, X_d$ and if the upper layers operations can be modelled by a polynomial $P$, then assuming the inputs $X_i$ have some degree of independence we could use this to modularly learn the lower and upper parts of the network separately (Appendix E)

---

[2]Here $H_4$ and $H_2$ are the fourth and second order hermite polynomials respectively.

[3]Theorem 1 holds for sigmoid with $t \geq c\log d$.

**Related Work.** Various works have attempted to understand the learnability of simple neural networks. Despite known hardness results Goel et al. (2016); Brutzkus & Globerson (2017), there has been an array of positive results under various distributional assumptions on the input and the underlying noise in the label. Most of these works have focused on analyzing one hidden layer neural networks. A line of research has focused on understanding the dynamics of gradient descent on these networks for recovering the underlying parameters under gaussian input distribution Du et al. (2017b;a); Li & Yuan (2017); Zhong et al. (2017a); Zhang et al. (2017); Zhong et al. (2017b). Another line of research borrows ideas from kernel methods and polynomial approximations to approximate the neural network by a linear function in a high dimensional space and subsequently learning the same Zhang et al. (2015); Goel et al. (2016); Goel & Klivans (2017b;a). Tensor decomposition methods Anandkumar & Ge (2016); Janzamin et al. (2015) have also been applied to learning these simple architectures.

The complexity of recovering arises from the highly non-convex nature of the loss function to be optimized. The main result we extend in this work is by Ge et al. (2017). They learn the neural network by designing a loss function that allows a "well-behaved" landscape for optimization avoiding the complexity. However, much like most other results, it is unclear how to extend to deeper networks. The only known result for networks with more than one hidden layer is by Goel & Klivans (2017b). Combining kernel methods with isotonic regression, they show that they can provably learn networks with sigmoids in the first hidden layer and a single unit in the second hidden layer in polynomial time. We however model the above layer as a multivariate polynomial allowing for larger representation. Another work Arora et al. (2014) deals with learning a deep generative network when several random examples are generated in an unsupervised setting. By looking at correlations between input coordinates they are able to recover the network layer by layer. We use some of their ideas in section 4 when $\mathbf{W}$ is a sparse binary matrix.

**Notation.** We denote vectors and matrices in bold face. $||\cdot||_p$ denotes the $l_p$-norm of a vector. $||\cdot||$ without subscript implies the $l_2$-norm. For matrices $||\cdot||$ denotes the spectral norm and $||\cdot||_F$ denotes the forbenius norm. $\mathcal{N}(0, \Sigma)$ denotes the multivariate gausssian distribution with mean 0 and covariance $\Sigma$. For a scalar $x$ we will use $\phi(x)$ to denote the p.d.f. of the univariate standard normal distribution with mean zero and variance 1 .For a vector $\boldsymbol{x}$ we will use $\phi(\boldsymbol{x})$ to denote the p.d.f. of the multivariate standard normal distribution with mean zero and variance 1 in each direction. $\Phi$ denotes the c.d.f. of the standard gausssian distribution. Also define $\Phi^c = 1 - \Phi$. Let $h_i$ denote the $i$th normalized Hermite polynomial Wikipedia contributors (2018). For a function $f$, let $\hat{f}_i$ denote the $i$th coefficient in the hermite expansion of $f$, that is, $\hat{f}_i = \mathbb{E}_{g \sim \mathcal{N}(0,1)}[f(g)h_i(g)]$. For a given function $f$ computed by the neural network, we assume that the training samples $(\boldsymbol{x}, y)$ are such that $\boldsymbol{x} \in \mathbb{R}^n$ is distributed according to $\mathcal{N}(0, 1)$ and label has no noise, that is, $y = f(\boldsymbol{x})$.

**Note:** Most proofs are deferred to the Appendix due to lack of space.

## 2 Approximate Recovery with Linear term

In this section we consider the case when $P$ has a positive linear component and we wish to recover the parameters of true parameters $\mathbf{W}^*$. The algorithm has two-steps: 1) uses existing one-hidden layer learning algorithm (SGD on carefully designed loss Ge et al. (2017)) to recover an approximate solution , 2) refine the approximate solution by performing local search (for monotone $P$). The intuition behind the first step is that high thresholds enable $P$ to in expectation be approximately close to a one-hidden-layer network which allows us to transfer algorithms with approximate guarantees. Secondly, with the approximate solutions as starting points, we can evaluate the closeness of the estimate of each weight vector to the true weight vector using simple correlations. The intuition of this step is to correlate with a function that is large only in the direction of the true weight vectors. This equips us with a way to design a local search based algorithm to refine the estimate to small error.

For simplicity in this section we will work with $P$ where the highest degree in any $X_i$ is 1. The degree of the overall polynomial can still be $n$. See Appendix B.8 for the extension to general $P$. More formally,

**Assumption 1** (Structure of network). *We assume that $P$ has the following structure $P(X_1, \ldots, X_k) = c_0 + \sum_{i \in [d]} c_i X_i + \sum_{S \subseteq [d]:|S|>1} c_S \prod_{j \in S} X_j$ such that $c_i = \Theta(1)^4$ for all $i \in [d]$ and for all $S \subseteq [d]$ such that $|S| > 1$, $|c_S| \leq O(1)$. $\mathbf{W}^*$ has constant condition number.*

Thus $f(\boldsymbol{x}) = c_0 + \sum_{i \in [d]} c_i u((\boldsymbol{w}_i^*)^T \boldsymbol{x}) + \sum_{S \subseteq [d]:|S|>1} c_S \prod_{j \in S} u((\boldsymbol{w}_j^*)^T \boldsymbol{x})$. Denote $f_{\text{lin}}(\boldsymbol{x}) = c_0 + \sum_{i \in [d]} c_i u((\boldsymbol{w}_i^*)^T \boldsymbol{x})$ to be the linear part of $f$.

Next we will upper bound expected value of $u(\boldsymbol{x})$: for "high-threshold" ReLU, that is, $u_t(a) = \max(0, a - t)$, $\mathbb{E}_{g \sim N(0,\sigma^2)}[u_t(g)]$ is bounded by a function $\rho(t, \sigma) \approx e^{-\frac{t^2}{2\sigma^2}}$ (see Lemma 10). We also get a lower bound on $|\hat{u}_4|$ in terms of $\rho(t, \sigma)$ [5] This enables us to make the following assumption.

**Assumption 2.** *Activation function $u$ is a positive high threshold activation with threshold $t$, that is, the bias term is $t$. $\mathbb{E}_{g \sim N(0,\sigma^2)}[u_t(g)] \leq \rho(t, \sigma)$ where $\rho$ is a positive decreasing function of $t$. Also, $|\hat{u}_k| = t^{\Theta(1)} \rho(t, 1)$ for $k = 2, 4$.*

**Assumption 3** (Value of $t$). *$t$ is large enough such that $\rho(t, ||\mathbf{W}^*||) \approx d^{-\eta}$ and $\rho(t, 1) \approx d^{-p\eta}$ with for large enough constant $\eta > 0$ and $p \in (0, 1]$.*

For example, for high threshold ReLU, $\rho(t, 1) = e^{-t^2/2}$ and $\mu = \rho(t, ||\mathbf{W}^*||) = e^{-t^2/2||\mathbf{W}^*||^2}$, thus $t = \sqrt{2\eta \log d}$ for large enough $d$ suffices to get the above assumption ($\kappa(\mathbf{W}^*)$ is a constant).

These high-threshold activation are useful for learning as in expectation, they ensure that $f$ is close to $f_{\text{lin}}$ since the product terms have low expected value. This is made clear by the following lemmas:

**Lemma 1.** *For $|S| > 1$, under Assumption 2 we have,*

$$\mathbb{E}\left[\prod_{j \in S} u_t((\boldsymbol{w}_j^*)^T \boldsymbol{x})\right] \leq \rho(t, 1) \left(\kappa(\mathbf{W}^*)\rho(t, ||\mathbf{W}^*||)\right)^{|S|-1}.$$

*So if $\mu := \kappa(\mathbf{W}^*)\rho(t, ||\mathbf{W}^*||)$, then $\mathbb{E}[\prod_{j \in S} X_j[\boldsymbol{x}]] \leq \rho(t, 1)\mu^{|S|-1}$*

**Lemma 2.** *Let $\Delta(\boldsymbol{x}) = f(\boldsymbol{x}) - f_{\text{lin}}(\boldsymbol{x})$. Under Assumptions 1, 2 and 3, if $t$ is such that $d\rho(t, ||\mathbf{W}^*||) \leq c$ for some small enough constant $c > 0$ we have,*

$$\mathbb{E}[|\Delta(\boldsymbol{x})|] \leq O\left(d^3 \rho(t, 1)\rho(t, ||\mathbf{W}^*||)\right) = O\left(d^{-(1+p)\eta+3}\right).$$

***Note:*** *We should point out that $f(\boldsymbol{x})$ and $f_{\text{lin}}(\boldsymbol{x})$ are very different point wise; they are just close in expectation under the distribution of $\boldsymbol{x}$. In fact, if $d$ is some constant then even the difference in expectation is some small constant.*

This closeness suggests that algorithms for recovering under the labels from $f_{\text{lin}}$ can be used to recover with labels from $f$ approximately.

**Learning One Layer Neural Networks using Landscape Design.** Ge et al. (2017) proposed an algorithm for learning one-hidden-layer networks. Intuitively, the approach of Ge et al. (2017) is to design a well behaved loss function based on correlations to recover the underlying weight vectors. They show that the local minima of the following optimization corresponds to some transform of each of the $\boldsymbol{w}_i^*$ – thus it can be used to recover a transform of $\boldsymbol{w}_i^*$, one at a time.

$$\max_{\boldsymbol{z}:\mathbb{E}[f_{\text{lin}}(\boldsymbol{x})H_2(\boldsymbol{z}^T \boldsymbol{x})]=\hat{u}_2} \text{sgn}(\hat{u}_4)\mathbb{E}[f_{\text{lin}}(\boldsymbol{x})H_4(\boldsymbol{z}^T \boldsymbol{x})]$$

which they optimize using the Lagrangian formulation (viewed as a minimization):

$$\min_{\boldsymbol{z}} G_{\text{lin}}(\boldsymbol{z}) := -\text{sgn}(\hat{u}_4)\mathbb{E}[f_{\text{lin}}(\boldsymbol{x})H_4(\boldsymbol{z}^T \boldsymbol{x})] + \lambda(\mathbb{E}[f_{\text{lin}}(\boldsymbol{x})H_2(\boldsymbol{z}^T \boldsymbol{x})] - \hat{u}_2)^2$$

where $H_2(\boldsymbol{z}^T \boldsymbol{x}) = ||\boldsymbol{z}||^2 h_2\left(\frac{\boldsymbol{z}^T \boldsymbol{x}}{||\boldsymbol{z}||}\right) = \frac{(\boldsymbol{z}^T \boldsymbol{x})^2}{\sqrt{2}} - \frac{||\boldsymbol{z}||^2}{\sqrt{2}}$ and $H_4(\boldsymbol{z}^T \boldsymbol{x}) = ||\boldsymbol{z}||^4 h_4\left(\frac{\boldsymbol{z}^T \boldsymbol{x}}{||\boldsymbol{z}||}\right) = \sqrt{6}\frac{(\boldsymbol{z}^T \boldsymbol{x})^4}{12} - \frac{||\boldsymbol{z}||^2(\boldsymbol{z}^T \boldsymbol{x})^2}{2} + \frac{||\boldsymbol{z}||^4}{4}$ (see Appendix A.1 for more details). Using properties

---

[4]We can handle $\in [d^{-C}, d^C]$ for some constant $C$ by changing the scaling on $t$.

[5]For similar bounds for sigmoid and sign refer to Appendix B.7.

of Hermite polynomials, we have $\mathbb{E}[f_{\text{lin}}(\boldsymbol{x})H_2(\boldsymbol{z}^T\boldsymbol{x})] = \hat{u}_2 \sum_i c_i(\boldsymbol{z}^T\boldsymbol{w}_i^*)^2$ and similarly $\mathbb{E}[f_{\text{lin}}(\boldsymbol{x})H_4(\boldsymbol{z}^T\boldsymbol{x})] = \hat{u}_4 \sum_i(\boldsymbol{z}^T\boldsymbol{w}_i^*)^4$. Thus

$$G_{\text{lin}}(\boldsymbol{z}) = -|\hat{u}_4| \sum_i c_i(\boldsymbol{z}^T\boldsymbol{w}_i^*)^4 + \lambda\hat{u}_2^2\left(\sum_i c_i(\boldsymbol{z}^T\boldsymbol{w}_i^*)^2 - 1\right)^2.$$

Using results from Ge et al. (2017), it can be shown that the approximate local minima of this problem are close to columns of $(\mathbf{TW}^*)^{-1}$ where $\mathbf{T}$ is a diagonal matrix with $T_{ii} = \sqrt{c_i}$.

**Definition 1** (($\epsilon, \tau$)-local minimum/maximum). *$\boldsymbol{z}$ is an ($\epsilon, \tau$)-local minimum of $F$ if $||\nabla F(\boldsymbol{z})|| \leq \epsilon$ and $\lambda_{\min}(\nabla^2 F(\boldsymbol{z})) \leq \tau$.*

**Claim 1** (Ge et al. (2017)). *An ($\epsilon, \tau$)-local minima of the Lagrangian formulation $\boldsymbol{z}$ with $\epsilon \leq O\left(\sqrt{\tau^3/|\hat{u}_4|}\right)$ is such that for an index $i$ $|\boldsymbol{z}^T\boldsymbol{w}_i| = 1 \pm O(\epsilon/\lambda\hat{u}_2^2) \pm O(d\tau/|\hat{u}_4|)$ and $\forall j \neq i$, $|v^T\boldsymbol{w}_j| = O(\sqrt{\tau/|\hat{u}_4|})$ where $\boldsymbol{w}_i$ are columns of $(\mathbf{TW}^*)^{-1}$.*

Ge et al. (2017) do not mention $\hat{u}_2$ but it is necessary in the non-orthogonal weight vectors case for the correct reduction. Since for us, this value can be small, we mention the dependence. Note that these are not exactly the directions $\boldsymbol{w}_i^*$ that we need, one way to think about is that we can get the correct directions by estimating all columns and then inverting.

**One-hidden-layer to Deep Neural Network.** Consider the loss with $f$ instead of $f_{\text{lin}}$:
$$\min \boldsymbol{z} : G(\boldsymbol{z}) = -\text{sgn}(\hat{u}_4)\mathbb{E}[f(\boldsymbol{x})H_4(\boldsymbol{z}^T\boldsymbol{x})] + \lambda(\mathbb{E}[f(\boldsymbol{x})H_2(\boldsymbol{z}^T\boldsymbol{x})] - \hat{u}_2)^2$$

We previously showed that $f$ is close to $f_{\text{lin}}$ in expectation due to the high threshold property. This also implies that $G_{\text{lin}}$ and $G$ are close and so are the gradients and (eigenvalues of) hessians of the same. This closeness implies that the landscape properties of one approximately transfers to the other function. More formally,

**Theorem 4.** *Let $\mathbf{Z}$ be an ($\epsilon, \tau$)-local minimum of function $A$. If $||\nabla(B-A)(\mathbf{Z})|| \leq \rho$ and $||\nabla^2(B-A)(\mathbf{Z})|| \leq \gamma$ then $\mathbf{Z}$ is an ($\epsilon + \rho, \tau + \gamma$)-local minimum of function $B$ and vice-versa.*

We will now apply above lemma on our $G_{\text{lin}}(\boldsymbol{z})$ and $G(\boldsymbol{z})$.

**Claim 2.** *For $\lambda = \Theta(|\hat{u}_4|/\hat{u}_2^2) \approx d^\eta$, an ($\epsilon, \tau$)-approximate local minima of $G$ (for small enough $\epsilon, \tau \leq d^{-2\eta}$) is an ($O(\log d)d^{-(1+p)\eta+3}, O(\log d)d^{-(1+p)\eta+3}$)-approximate local minima of $G_{\text{lin}}$. This implies $\boldsymbol{z}$ is such that for an index $i$, $|\boldsymbol{z}^T\boldsymbol{w}_i| = 1 \pm O(1)d^{-2/3p\eta+3}$ and $\forall j \neq i$, $|\boldsymbol{z}^T\boldsymbol{w}_j| = O(1)d^{-1/3p\eta+3/2}$ where $\boldsymbol{w}_i$ are columns of $(\mathbf{TW}^*)^{-1}$ (ignoring $\log d$ factors).*

**Note:** For ReLU, setting $t = \sqrt{C\log d}$ for large enough $C > 0$ we can get closeness $1/\text{poly}(d)$ to the columns of $(\mathbf{TW}^*)^{-1}$. Refer Appendix B.7 for details for sigmoid.

The paper Ge et al. (2017) also provides an alternate optimization that when minimized simultaneously recovers the entire matrix $\mathbf{W}^*$ instead of having to learn columns of $(\mathbf{TW}^*)^{-1}$ separately. We show how applying our methods can also be applied to that optimization in Appendix B.4 to recover $\mathbf{W}^*$ by optimizing a single objective.

## 2.1 APPROXIMATE TO ARBITRARILY CLOSE FOR MONOTONE $P$

Assuming $P$ is monotone, we can show that the approximate solution from the previous analysis can be refined to arbitrarily closeness using a random search method followed by approximately finding the angle of our current estimate to the true direction.

The idea at a high level is to correlate with $\delta'(\boldsymbol{z}^T\boldsymbol{x} - t)$ where $\delta$ is the Dirac delta function. It turns out that the correlation is maximized when $z$ is equal to one of the $w_i$. Correlation with $\delta'(\boldsymbol{z}^T\boldsymbol{x}-t)$ is checking how fast the correlation of $f$ with $\delta(\boldsymbol{z}^T\boldsymbol{x}-t)$ is changing as you change $t$. To understand this look at the case when our activation $u$ is the sign function then note that correlation of $u_t(\boldsymbol{w}^T\boldsymbol{x} - t)$ with $\delta'(\boldsymbol{w}^T\boldsymbol{x} - t)$ is very high as its correlation with $\delta(\boldsymbol{w}^T\boldsymbol{x} - t')$ is 0 when $t' < t$ and significant when $t' > t$. So as we change t' slightly from $t-\epsilon$ to $t+\epsilon$ there is a sudden increase. If $z$ and $w$ differ then it can be shown that correlation of $u_t(\boldsymbol{w}^T\boldsymbol{x} - t)$ with $\delta'(\boldsymbol{z}^T\boldsymbol{x} - t)$ essentially depends on $\cot(\alpha)$ where $\alpha$ is the angle between $w$ and $z$ (for a quick intuition note that one can

prove that $\mathbb{E}[u_t(\boldsymbol{w}^T\boldsymbol{x})\delta'(\boldsymbol{z}^T\boldsymbol{x})] = c\cot(\alpha)$. See Lemma 16 in Appendix). In the next section we will show how the same ideas work for non-monotone $P$ even if it may not have any linear terms but we only manage to prove polynomial sample complexity for finding $w$ instead of polynomial time complexity.

In this section we will not correlate exactly with $\delta'(\boldsymbol{z}^T\boldsymbol{x} - t)$ but instead we will use this high level idea to estimate how fast the correlation with $\delta(\boldsymbol{z}^T\boldsymbol{x} - t')$ changes between two specific values as one changes $t'$, to get an estimate for $cot(\alpha)$. Secondly since we can't to a smooth optimization over $z$, we will do a local search by using a random perturbation and iteratively check if the correlation has increased. We can assume that the polynomial $P$ doesn't have a constant term $c_0$ as otherwise it can easily be determined and cancelled out[6].

We will refine the weights one by one. WLOG, let us assume that $\boldsymbol{w}_1^* = \boldsymbol{e}_1$ and we have $\boldsymbol{z}$ such that $\boldsymbol{z}^T\boldsymbol{w}_1^* = z_1 = \cos^{-1}(\alpha_1)$. Let $l(\boldsymbol{z}, t, \epsilon)$ denote $\{\boldsymbol{x} : \boldsymbol{z}^T\boldsymbol{x} \in [t - \epsilon, t]\}$ for $\boldsymbol{z} \in S^{n-1}$.

---

**Algorithm 1** RefineEstimate

---

1: Run $EstimateTanAlpha$ on $\boldsymbol{z}$ to get $s = \tan(\alpha)$ where $\alpha$ is the angle between $\boldsymbol{z}$ and $w_1^*$.
2: Perturb current estimate $\boldsymbol{z}$ by a vector along the $d-1$ dimensional hyperplane normal to $\boldsymbol{z}$ with the distribution $n(0, \Theta(\alpha/d))^{d-1}$ to get $\boldsymbol{z}'$.
3: Run $EstimateTanAlpha$ on $\boldsymbol{z}'$ to get $s' = \tan(\alpha')$ where $\alpha'$ is the angle between $\boldsymbol{z}'$ and $w_1^*$.
4: **if** $\alpha' \leq O(\alpha/d)$ **then**
5: $\quad \boldsymbol{z} \leftarrow \boldsymbol{z}'$
6: Repeat till $\alpha' \leq \epsilon$.

---

---

**Algorithm 2** EstimateTanAlpha

---

1: Find $t_1$ and $t_2$ such that $Pr[\text{sgn}(f(\boldsymbol{x}))|\boldsymbol{x} \in l(\boldsymbol{z}, t', \epsilon)]$ at $t_1$ is 0.4 and at $t_2$ is 0.6.
2: Return $\frac{t_2 - t_1}{\Phi^{-1}(0.6) - \Phi^{-1}(0.4)}$.

---

The algorithm (Algorithm 1) estimates the angle of the current estimate with the true vector and then subsequently perturbs the vector to get closer after each successful iteration.

**Theorem 5.** *Given a vector $\boldsymbol{z} \in \mathbb{S}^{d-1}$ such that it is $1/\text{poly}(d)$-close to the underlying true vector $\boldsymbol{w}_1^*$, that is $\cos^{-1}(\boldsymbol{z}^T\boldsymbol{w}_1^*) \leq 1/\text{poly}(d)$, running $RefineEstimate$ for $O(T)$ iterations outputs a vector $\boldsymbol{z}^* \in \mathbb{S}^{d-1}$ such that $\cos^{-1}((\boldsymbol{z}^*)^T\boldsymbol{w}_1^*) \leq \left(1 - \frac{c}{d}\right)^T \gamma$ for some constant $c > 0$. Thus after $O(d\log(1/\epsilon))$ iterations $\cos^{-1}((\boldsymbol{z}^*)^T\boldsymbol{w}_1^*) \leq \epsilon$.*

We prove the correctness of the algorithm by first showing that $EstimateTanAlpha$ gives a multiplicative approximation to $\tan(\alpha)$. The following lemma captures this property.

**Lemma 3.** *$EstimateTanAlpha(\boldsymbol{z})$ outputs $y$ such that $y = (1 \pm O(\eta))\tan(\alpha)$ where $\alpha$ is the angle between $\boldsymbol{z}$ and $w_1^*$.*

*Proof.* We first show that the given probability when computed with $\text{sgn}(\boldsymbol{x}^T\boldsymbol{w}_1^* - t)$ is a well defined function of the angle between the current estimate and the true parameter up to multiplicative error. Subsequently we show that the computed probability is close to the one we can estimate using $f(\boldsymbol{x})$ since the current estimate is close to one direction. The following two lemmas capture these properties.

**Lemma 4.** *For $t, t'$ and $\epsilon \leq 1/t'$, we have*

$$Pr[\boldsymbol{x}^T\boldsymbol{w}_1^* \geq t \text{ and } \boldsymbol{x} \in l(\boldsymbol{z}, t', \epsilon)|\boldsymbol{x} \in l(\boldsymbol{z}, t, \epsilon)] = \Phi^c\left(\frac{t - t^*\cos(\alpha_1)}{|\sin(\alpha_1)|}\right) \pm O(\epsilon)t'$$

**Lemma 5.** *For $t' \in [0, t/\cos(\alpha_1)]$, we have*

$$Pr[sgn(f(\boldsymbol{x}))|\boldsymbol{x} \in l(\boldsymbol{z}, t', \epsilon)] = Pr[sgn((\boldsymbol{w}_1^*)^T\boldsymbol{x} - t)|\boldsymbol{x} \in l(\boldsymbol{z}, t, \epsilon)] + de^{-\Omega(t^2)}.$$

---

[6]for example with RELU activation, $f$ will be $c_0$ most of the time as other terms in $P$ will never activate. So $c_0$ can be set to say the median value of $f$.

Using the above, we can show that,

$$t_2 - t_1 = \left( \Phi^{-1}(0.6 - \eta_1 \pm O(\epsilon)t_1) - \Phi^{-1}(0.4 - \eta_2 \pm O(\epsilon)t_2) \right) \tan(\alpha)$$

$$= \left( \Phi^{-1}(0.6) - \Phi^{-1}(0.4) - (\eta_1 \pm O(\epsilon)t_1)(\Phi^{-1})'(p_1) + (\eta_2 \pm O(\epsilon)t_2)(\Phi^{-1})'(p_2) \right) \tan(\alpha)$$

where $\eta_1, \eta_2 > 0$ are the noise due to estimating using $f$ and $p_1 \in [0.6 - \eta_1 \pm O(\epsilon)t_1, 0.6]$ and $p_2 \in [0.4 - \eta_2 \pm O(\epsilon)t_2, 0.4]$ as long as $t_1, t_2 \in [0, t/\cos(\alpha_1)]$. The following lemma bounds the range of $t_1$ and $t_2$.

**Lemma 6.** *We have* $0 \le t_1 \le t_2 \le \frac{t}{\cos(\alpha_1)}$.

Thus, we have,

$$\frac{t_2 - t_1}{\Phi^{-1}(0.6) - \Phi^{-1}(0.4)} = (1 \pm O(\eta_1 + \eta_2 + \epsilon t_2)) \tan(\alpha)$$

as long as $\eta_2 + O(\epsilon)t_2 \le c$ for some constant $c > 0$. Thus, we can get a multiplicative approximation to $\tan(\alpha)$ up to error $\eta$ ($\epsilon$ can be chosen to make its contribution smaller than $\eta$).

Finally we show (proof in Appendix **??**) that with constant probability, a random perturbation reduces the angle by a factor of $(1 - 1/d)$ of the current estimate hence the algorithm will halt after $O(d \log(1/\nu))$ iterations.

**Lemma 7.** *By applying a random Gaussian perturbation along the $d - 1$ dimensional hyperplane normal to $\mathbf{z}$ with the distribution $n(0, \Theta(\alpha/d))^{d-1}$ and scaling back to the unit sphere, with constant probability, the angle $\alpha$ $(< \pi/2)$ with the fixed vector decreases by at least $\Omega(\alpha/d)$.*

$\square$

## 3 SAMPLE COMPLEXITY

We extend the methods of the previous section to a broader class of polynomials but only to obtain results in terms of sample complexity. The main idea as in the previous section is to correlate with $\delta'(\mathbf{z}^T\mathbf{x} - t)$ (the derivative of the dirac delta function) and find $\arg\max_{||\mathbf{z}||_2=1} \mathbb{E}[f(\mathbf{x})\delta'(\mathbf{z}^T\mathbf{x} - t)]$. We will show that the correlation goes to infinity when $\mathbf{z}$ is one of $\mathbf{w}_i^*$ and bounded if it is far from all of them. From a practical standpoint we calculate $\delta'(\mathbf{z}^T\mathbf{x} - s)$ by measuring correlation with $\frac{1}{2\epsilon}(\delta(\mathbf{z}^T\mathbf{x} - s + \epsilon) - \delta(\mathbf{z}^T\mathbf{x} - s - \epsilon))$. In the limit as $\epsilon \to 0$ this becomes $\delta'(\mathbf{z}^T\mathbf{x} - s)$. $\delta(\mathbf{z}^T\mathbf{x} - s)$ in turn is estimated using $\frac{1}{\epsilon}(\text{sgn}(\mathbf{z}^T\mathbf{x} - s + \epsilon) - \text{sgn}(\mathbf{z}^T\mathbf{x} - s))$, as in the previous section, for an even smaller $\epsilon$; however, for ease of exposition, in this section, we will assume that correlations with $\delta(\mathbf{z}^T\mathbf{x} - s)$ can be measured exactly.

Let us recall that $f(\mathbf{x}) = P(u((\mathbf{w}_1^*)^T\mathbf{x}), u((\mathbf{w}_2^*)^T\mathbf{x}), \ldots, u((\mathbf{w}_d^*)^T\mathbf{x}))$. Let $C_1(f, \mathbf{z}, s)$ denote $E[f(\mathbf{x})\delta(\mathbf{z}^T\mathbf{x} - s)]$ and let $C_2(f, \mathbf{z}, s)$ denote $E[f(\mathbf{x})(\delta(\mathbf{z}^T\mathbf{x} - s - \epsilon) - \delta(\mathbf{z}^T\mathbf{x} - s + \epsilon)]$.

If $u = \text{sgn}$ then $P$ has degree at most 1 in each $X_i$. Let $\frac{\partial P}{\partial X_i}$ denote the symbolic partial derivative of $P$ with respect to $X_i$; so, it drops monomials without $X_i$ and factors off $X_i$ from the remaining ones. Let us separate dependence on $X_i$ in $P$ as follows:

$$P(X_1, , .., X_d) = X_i Q_i(X_1, ..X_{i-1}, X_{i+1}, .., X_d) + R_1(X_1, .X_{i-1}, X_{i+1}, .., X_d)$$

then $\frac{\partial P}{\partial X_i} = Q_i$.

We will overload the polynomial $P$ such that $P[\mathbf{x}]$ to denote the polynomial computed by substituting $X_i = u((\mathbf{w}_1^*)^T\mathbf{x})$ and similarly for $Q$ and $R$. Under this notation $f(\mathbf{x}) = P[\mathbf{x}]$. We will also assume that $|P(X)| \le ||X||^{O(1)} = ||X||^{c_1}$ (say). By using simple correlations we will show:

**Theorem 6.** *If $u$ is the sgn function, $P(X) \le ||X||^{c_1}$ and for all $i$, $E[Q_i[\mathbf{x}]|(\mathbf{w}_i^*)^T\mathbf{x} = t] \ge \epsilon_3$ then using $\text{poly}(\frac{d}{\epsilon_3\epsilon_2})$ samples one can determine the $\mathbf{w}_i^*$'s within error $\epsilon_2$.[7]*

Note that if all the $\mathbf{w}_i^*$'s are orthogonal then $X_i$ are independent and $E\left[Q_i[\mathbf{x}]|(\mathbf{w}_i^*)^T\mathbf{x} = t\right]$ is just value of $Q_i$ evaluated by setting $X_i = 1$ and setting all the the remaining $X_j = \mu$ where $\mu = E[X_j]$. This is same as $1/\mu$ times the coefficient of $X_i$ in $P(\mu(X_1 + 1), \ldots, \mu(X_d + 1))$.

---

[7]The theorem can be extended to ReLU by correlating with the second derivative $\delta''$ (see Appendix C.1).

**Corollary 1.** *If $u$ is the sgn function and $\boldsymbol{w}_i^*$s are orthogonal then in sample complexity* $\text{poly}(\frac{d}{\epsilon_3 \epsilon_2})$ *one can determine $\mathbf{W}^*$ within error $\epsilon_2$ in each entry, if the coefficient of the linear terms in $P(\mu(X_1 + 1), \mu(X_2 + 1), \mu(X_3 + 1), ..)$ is larger than $\epsilon_3 \mu$, where $\mu = E[X_i]$.*

The main point behind the proof of Theorem 6 is that the correlation is high when $\boldsymbol{z}$ is along one of $\boldsymbol{w}_i^*$ and negligible if it is not close to any of them.

**Lemma 8.** *Assuming $P(X) < ||X||^{c_1}$. If $\boldsymbol{z} = \boldsymbol{w}_i^*$ then $C_2(f, \boldsymbol{z}, t) = \phi(t) E\left[\frac{\partial P}{\partial X_i} \Big| \boldsymbol{z}^T \boldsymbol{x} = t\right] + \epsilon d^{O(1)}$. Otherwise if all angles $\alpha_i$ between $\boldsymbol{z}$ and $\boldsymbol{w}_i^*$ are at least $\epsilon_2$ it is at most $\epsilon d^{O(1)}/\epsilon_2$.*

We will use the notation $g(x)_{x=s}$ to denote $g(x)$ evaluated at $x = s$. Thus Cauchy's mean value theorem can be stated as $g(x + \epsilon) - g(x) = \epsilon[g'(s)](s = s' \in [x, x + \epsilon])$. We will over load the notation a bit: $\phi(\boldsymbol{z}^T \boldsymbol{x} = s)$ will denote the probability density that $v z^T \boldsymbol{x} = s$; so if $\boldsymbol{z}$ is a unit vector this is just $\phi(s)$; $\phi(\boldsymbol{z}_1^T \boldsymbol{x} = s_1, \boldsymbol{z}_2^T \boldsymbol{x} = s_2)$ denotes the probability density that both $\boldsymbol{z}_1^T \boldsymbol{x} = s_1, \boldsymbol{z}_2^T \boldsymbol{x} = s_2$; so again if $\boldsymbol{z}_1, \boldsymbol{z}_2$ are orthonormal then this is just $\phi(s_1)\phi(s_2)$.

The following claim interprets correlation with $\delta(\boldsymbol{z}^T \boldsymbol{x} - s)$ as the expected value along the corresponding plane $\boldsymbol{z}^T \boldsymbol{x} = s$.

**Claim 3.** $E[f(\boldsymbol{x})\delta(\boldsymbol{z}^T \boldsymbol{x} - s)] = E[f(\boldsymbol{x})|\boldsymbol{z}^T \boldsymbol{x} = s]\phi(\boldsymbol{z}^T \boldsymbol{x} = s)$.

The following claim computes the correlation of $P$ with $\delta'(\boldsymbol{z}^T \boldsymbol{x} - s)$.

**Claim 4.** $\mathbb{E}[P[\boldsymbol{x}]\delta'(\boldsymbol{z}^T \boldsymbol{x} = s)]$ *is equal to* $\sum_i |\cot(\alpha_i)|\phi(\boldsymbol{z}^T \boldsymbol{x} = s, (\boldsymbol{w}_i^*)^T \boldsymbol{x} = t)$ $\mathbb{E}\left[\frac{\partial P}{\partial X_i}[\boldsymbol{x}]|\boldsymbol{z}^T \boldsymbol{x} = s, (\boldsymbol{w}_i^*)^T \boldsymbol{x} = t\right] + \phi'(s)E[P[\boldsymbol{x}]|\boldsymbol{z}^T \boldsymbol{x} = s]$.

We use this to show that the correlation is bounded if all the angles are lower bounded.

**Claim 5.** *If $P(X) \leq ||X||^{c_1}$ and if $\boldsymbol{z}$ has an angle of at least $\epsilon_2$ with all the $\boldsymbol{w}_i^*$'s then $C_2(f, \boldsymbol{z}, s) \leq \epsilon d^{O(1)}/\epsilon_2$.*

Above claims can be used to prove main Lemma 8. Refer to the Appendix C for proofs.

*Proof of Theorem 6.* If we wish to determine $\boldsymbol{w}_i^*$ within an angle of accuracy $\epsilon_2$ let us set $\epsilon$ to be $O(\epsilon_3 \epsilon_2 \phi(t) d^{-c})$. From Lemma 8, for some large enough $c$, this will ensure that if all $\alpha_i > \epsilon_2$ the correlation is $o(\phi(t)\epsilon_3)$. Otherwise it is $\phi(t)\epsilon_3(1 \pm o(1))$. Since $\phi(t) = poly(1/d)$, given $poly(\frac{d}{\epsilon_2 \epsilon_3})$ samples, we can test if a given direction is within accuracy $\epsilon_2$ of a $\boldsymbol{w}_i^*$ or not. $\square$

## 4 STRONGER RESULTS UNDER STRUCTURAL ASSUMPTIONS

Under additional structural assumptions on $\mathbf{W}^*$ such as the weights being binary, that is, in $\{0, 1\}$, sparsity or certain restrictions on activation functions, we can give stronger recovery guarantees. Proofs have been deferred to Appendix D.

**Theorem 7.** *For activation $u_t(a) = e^{\rho(a-t)}$. Let the weight vectors $\boldsymbol{w}_i^*$ be $0, 1$ vectors that select the coordinates of $\boldsymbol{x}$. For each $i$, there are exactly $d$ indices $j$ such that $\boldsymbol{w}_{ij} = 1$ and the coefficient of the linear terms in $P(\mu(X_1 + 1), \mu(X_2 + 1), \mu(X_3 + 1), ..)$ for $\mu = e^{-\rho t}$ is larger than the coefficient of all the product terms (constant factor gap) then we can learn the $\mathbf{W}^*$.*

In order to prove the above, we will construct a correlation graph over $x_1, \ldots, x_n$ and subsequently identify cliques in the graph to recover $\boldsymbol{w}_i^*$'s.

With no threshold, recovery is still possible for disjoint, low $l_1$-norm vector. The proof uses simple correlations and shows that the optimization landscape for maximizing these correlations has local maximas being $\boldsymbol{w}_i^*$'s.

**Theorem 8.** *For activation $u(a) = e^a$. If all $\boldsymbol{w}_i^* \in \{0, 1\}^n$ are disjoint, then we can learn $\boldsymbol{w}_i^*$ as long as $P$ has all positive coefficients and product terms have degree at most 1 in each variable.*

For even activations, it is possible to recover the weight vectors even when the threshold is 0. The technique used is the PCA like optimization using hermite polynomials as in Section 2. Denote $C(S, \mu) = \sum_{S \subseteq S' \subseteq [n]} c_{S'} \mu^{|S'|}$.

**Theorem 9.** *If the activation is even and for every $i, j$: $C(\{i\}, \hat{u}_0) + C(\{j\}, \hat{u}_0) > \frac{6\hat{u}_2^2}{\hat{u}_0 \hat{u}_4} C(\{i, j\}, \hat{u}_0)$ then there exists an algorithm that can recover the underlying weight vectors.*

## 5 CONCLUSION

In this work we show how activations in a deep network that have a high threshold make it easier to learn the lowest layer of the network. We show that for a large class of functions that represent the upper layers, the lowest layer can be learned with high precision. Even if the threshold is low we show that the sample complexity is polynomially bounded. An interesting open direction is to apply these methods to learn all layers recursively. It would also be interesting to obtain stronger results if the high thresholds are only present at a higher layer based on the intuition we discussed.

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

## A    PREREQUISITES

### A.1    HERMITE POLYNOMIALS

Hermite polynomials form a complete orthogonal basis for the gaussian distribution with unit variance. For more details refer to Wikipedia contributors (2018). Let $h_i$ be the normalized hermite polynomials. They satisfy the following,

**Fact 0.** $\mathbb{E}[h_n(x)] = 0$ for $n > 0$ and $\mathbb{E}[h_0(x)] = 1$.

**Fact 1.** $\mathbb{E}_{a \sim N(0,1)}[h_i(a)h_j(a)] = \delta_{ij}$ where $\delta_{ij} = 1$ iff $i = j$.

This can be extended to the following:

**Fact 2.** For $a, b$ with marginal distribution $N(0, 1)$ and correlation $\rho$, $\mathbb{E}[h_i(a)h_j(b)] = \delta_{ij}\rho^j$.

Consider the following expansion of $u$ into the hermite basis $(h_i)$,

$$u(a) = \sum_{i=0}^{\infty} \hat{u}_i h_i(a).$$

**Lemma 9.** *For unit norm vectors* $u, v$, $\mathbb{E}[u(\boldsymbol{v}^T\boldsymbol{x})h_j(\boldsymbol{w}^T\boldsymbol{x})] = \hat{u}_j(\boldsymbol{v}^T\boldsymbol{w})^j$.

*Proof.* Observe that $\boldsymbol{v}^T\boldsymbol{x}$ and $\boldsymbol{w}^T\boldsymbol{x}$ have marginal distribution $N(0, 1)$ and correlation $\boldsymbol{v}^T\boldsymbol{w}$. Thus using Fact 2,

$$\mathbb{E}[u(\boldsymbol{v}^T\boldsymbol{x})h_j(\boldsymbol{w}^T\boldsymbol{x})] = \sum_{i=1}^{\infty} \hat{u}_i \mathbb{E}[h_i(\boldsymbol{v}^T\boldsymbol{x})h_j(\boldsymbol{w}^T\boldsymbol{x})] = \sum_{i=1}^{\infty} \hat{u}_i \delta_{ij}(\boldsymbol{v}^T\boldsymbol{w})^j = \hat{u}_j(\boldsymbol{v}^T\boldsymbol{w})^j.$$

$\square$

For gaussians with mean 0 and variance $\sigma^2$ define weighted hermite polynomials $H_l^{\sigma}(a) = |\sigma|^l h_l(a/\sigma)$. Given input $\boldsymbol{v}^T\boldsymbol{x}$ for $\boldsymbol{x} \sim N(0, \mathbf{I})$, we suppress the superscript $\sigma = ||\boldsymbol{v}||$.

**Corollary 2.** *For a non-zero vector* $\boldsymbol{v}$ *(not necessarily unit norm) and a unit norm vector* $\boldsymbol{w}$, $\mathbb{E}[H_i(\boldsymbol{v}^T\boldsymbol{x})h_j(\boldsymbol{w}^T\boldsymbol{x})] = \delta_{ij}(\boldsymbol{v}^T\boldsymbol{w})^j$.

*Proof.* It follows as the proof of the previous lemma,

$$\mathbb{E}[u(\boldsymbol{v}^T\boldsymbol{x})h_j(\boldsymbol{w}^T\boldsymbol{x})] = \sum_{i=1}^{\infty} \hat{u}_i \mathbb{E}[h_i(\boldsymbol{v}^T\boldsymbol{x})h_j(\boldsymbol{w}^T\boldsymbol{x})] = \sum_{i=1}^{\infty} \hat{u}_i \delta_{ij}(\boldsymbol{v}^T\boldsymbol{w})^j = \hat{u}_j(\boldsymbol{v}^T\boldsymbol{w})^j.$$

$\square$

**Fact 3.** $h_n(x + y) = 2^{-\frac{n}{2}} \sum_{k=0}^{n} \binom{n}{k} h_{n-k}(x\sqrt{2})h_k(y\sqrt{2})$.

**Fact 4.** $h_n(\gamma x) = \sum_{k=0}^{\lfloor \frac{n}{2} \rfloor} \gamma^{n-2k}(\gamma^2 - 1)^k \binom{n}{2k} \frac{(2k)!}{k!} 2^{-k} h_{n-2k}(x)$.

**Fact 5.** $\alpha(n, m, \gamma) = \mathbb{E}[h_m(x)h_n(\gamma x)] = \gamma^{n-2k}(\gamma^2 - 1)^k \binom{n}{2k} \frac{(2k)!}{k!} 2^{-k}$ for $k = \frac{n-m}{2}$ if $k \in \mathbb{Z}^+$ else 0.

### A.2    PROPERTIES OF MATRICES

Consider matrix $\mathbf{A} \in \mathbb{R}^{m \times m}$. Let $\sigma_i(\mathbf{A})$ to be the $i$th singular value of $A$ such that $\sigma_1(\mathbf{A}) \geq \sigma_2(\mathbf{A}) \geq \ldots \geq \sigma_m(\mathbf{A})$ and set $\kappa(\mathbf{A}) = \sigma_1(\mathbf{A})/\sigma_m(\mathbf{A})$.

**Fact 6.** $|\det(\mathbf{A})| = \prod_{i=1}^{m} \sigma_i(\mathbf{A})$.

**Fact 7.** Let $B$ be a $(mk) \times (mk)$ principal submatrix of $A$, then $\kappa(\mathbf{B}) \leq \kappa(\mathbf{A})$.

## A.3 ACTIVATION FUNCTIONS

**Lemma 10.** *For $u$ being a high threshold ReLU, that is, $u_t(a) = \max(0, a - t)$ we have for $t \geq C$ for large enough constant $C > 0$, $\mathbb{E}_{g \sim N(0,\sigma^2)}[u_t(g)] \leq e^{-\frac{t^2}{2\sigma^2}}$. Also, $\hat{u}_4, \hat{u}_2 = t^{\Theta(1)} e^{-\frac{t^2}{2}}$.*

*Proof.* We have

$$
\begin{aligned}
\mathbb{E}_{g \sim N(0,\sigma^2)}[u_t(g)] &= \frac{1}{\sqrt{2\pi}\sigma} \int_{-\infty}^{\infty} \max(0, g - t) e^{-\frac{g^2}{2\sigma^2}} dg \\
&= \frac{1}{\sqrt{2\pi}\sigma} \int_{t}^{\infty} (g - t) e^{-\frac{g^2}{2\sigma^2}} dg \\
&\leq \frac{1}{\sqrt{2\pi}\sigma} \int_{t}^{\infty} g e^{-\frac{g^2}{2\sigma^2}} dg \\
&= \frac{\sigma}{\sqrt{2\pi}} \int_{\frac{t^2}{2\sigma^2}}^{\infty} e^{-h} dh \\
&= \frac{\sigma}{\sqrt{2\pi}} e^{-\frac{t^2}{2\sigma^2}}.
\end{aligned}
$$

Also,

$$
\begin{aligned}
\hat{u}_4 &= \mathbb{E}_{g \sim N(0,1)}[u_t(g) h_4(g)] \\
&= \frac{1}{\sqrt{2\pi}} \int_{-\infty}^{\infty} \max(0, g - t)(g^4 - 6g^2 + 3) e^{-\frac{g^2}{2}} dg \\
&= \frac{1}{\sqrt{2\pi}} \int_{t}^{\infty} (g - t)(g^4 - 6g^2 + 3) e^{-\frac{g^2}{2}} dg \\
&\geq \frac{1}{\sqrt{2\pi}} (t^4 - 6t^2) \frac{1}{t} e^{-\frac{t^2}{2} - 1 - \frac{1}{2t^2}} \\
&\geq \Omega\left(t^3 e^{-\frac{t^2}{2}}\right).
\end{aligned}
$$

To upper bound,

$$
\begin{aligned}
\hat{u}_4 &= \frac{1}{\sqrt{2\pi}} \int_{-\infty}^{\infty} \max(0, g - t)(g^4 - 6g^2 + 3) e^{-\frac{g^2}{2}} dg \\
&= \frac{1}{\sqrt{2\pi}} \int_{t}^{\infty} (g - t)(g^4 - 6g^2 + 3) e^{-\frac{g^2}{2}} dg \\
&\leq \frac{1}{\sqrt{2\pi}} \int_{t}^{\infty} 2g^5 e^{-\frac{g^2}{2}} dg \\
&= \frac{1}{\sqrt{2\pi}} \int_{\frac{t^2}{2}}^{\infty} h^2 e^{-h} dh \\
&= O\left(t^4 e^{-\frac{t^2}{2}}\right).
\end{aligned}
$$

Similar analysis holds for $\hat{u}_2$. $\square$

Observe that sgn can be bounded very similarly replacing $g - t$ by 1 which can affect the bounds up to only a polynomial in $t$ factor.

**Lemma 11.** *For $u$ being a high threshold sgn, that is, $u_t(a) = sgn(a - t)$ we have for $t \geq C$ for large enough constant $C > 0$, $\mathbb{E}_{g \sim N(0,\sigma^2)}[u_t(g)] \leq e^{-\frac{t^2}{2\sigma^2}}$. Also, $\hat{u}_4, \hat{u}_2 = t^{\Theta(1)} e^{-\frac{t^2}{2}}$.*

For sigmoid, the dependence varies as follows:

**Lemma 12.** *For $u$ being a high threshold sigmoid, that is, $u_t(a) = \frac{1}{1 + e^{-(a-t)}}$ we have for $t \geq C$ for large enough constant $C > 0$, $\mathbb{E}_{g \sim N(0,\sigma^2)}[u_t(g)] \leq e^{-t + \frac{\sigma^2}{2}}$. Also, $\hat{u}_4, \hat{u}_2 = \Theta(e^{-t})$.*

*Proof.* We have

$$
\begin{aligned}
\mathbb{E}_{g \sim N(0,\sigma^2)}[u_t(g)] &= \frac{1}{\sqrt{2\pi}\sigma} \int_{-\infty}^{\infty} \frac{1}{1 + e^{-(g-t)}} e^{-\frac{g^2}{2\sigma^2}} \, dg \\
&= \frac{e^{-t}}{\sqrt{2\pi}\sigma} \int_{-\infty}^{\infty} \frac{1}{e^{-t} + e^{-g}} e^{-\frac{g^2}{2\sigma^2}} \, dg \\
&\leq \frac{e^{-t}}{\sqrt{2\pi}\sigma} \int_{-\infty}^{\infty} e^g e^{-\frac{g^2}{2\sigma^2}} \, dg \\
&= \frac{e^{-t} e^{\frac{\sigma^2}{2}}}{\sqrt{2\pi}\sigma} \int_{-\infty}^{\infty} e^{-\frac{(g-\sigma^2)^2}{2\sigma^2}} \, dg \\
&= e^{-t} e^{\frac{\sigma^2}{2}}
\end{aligned}
$$

Also,

$$
\begin{aligned}
\hat{u}_4 &= \mathbb{E}_{g \sim N(0,1)}[u_t(g)h_4(g)] \\
&= \frac{1}{\sqrt{2\pi}} \int_{-\infty}^{\infty} \frac{1}{1 + e^{-(g-t)}} e^{-\frac{g^2}{2}} \, dg \\
&= \frac{e^{-t}}{\sqrt{2\pi}} \int_{-\infty}^{\infty} \frac{1}{e^{-t} + e^{-g}} (g^4 - 6g^2 + 3) e^{-\frac{g^2}{2}} \, dg \\
&\geq \frac{e^{-t}}{\sqrt{2\pi}} \int_{0}^{\infty} \frac{1}{e^{-t} + e^{-g}} (g^4 - 6g^2 + 3) e^{-\frac{g^2}{2}} \, dg \\
&\geq \frac{e^{-t}}{\sqrt{2\pi}} \int_{0}^{\infty} \frac{1}{2} (g^4 - 6g^2 + 3) e^{-\frac{g^2}{2}} \, dg \\
&= \Omega(e^{-t}).
\end{aligned}
$$

$\square$

We can upper bound similarly and bound $\hat{u}_2$.

# B  APPROXIMATE RECOVERY WITH LINEAR TERMS

## B.1  CONSTRAINED OPTIMIZATION VIEW OF LANDSCAPE DESIGN

Let us consider the linear case with $\boldsymbol{w}_i^*$'s are orthonormal. Consider the following maximization problem for even $l \geq 4$,

$$
\max_{\boldsymbol{z} \in S^{n-1}} \operatorname{sgn}(\hat{u}_l) \cdot \mathbb{E}\left[ f(\boldsymbol{x}) \cdot H_l\left(\boldsymbol{z}^T \boldsymbol{x}\right) \right]
$$

where $h_l$ is the $l$th hermite polynomial. Then we have,

$$
\begin{aligned}
\operatorname{sgn}(\hat{u}_l) \cdot \mathbb{E}\left[ f(\boldsymbol{x}) \cdot h_l\left(\boldsymbol{z}^T \boldsymbol{x}\right) \right] &= \operatorname{sgn}(\hat{u}_l) \cdot \mathbb{E}\left[ \left( \sum_{i=1}^{k} c_i u_t((\boldsymbol{w}_i^*)^T \boldsymbol{x}) \right) \cdot h_l\left(\boldsymbol{z}^T \boldsymbol{x}\right) \right] \\
&= \operatorname{sgn}(\hat{u}_l) \cdot \sum_{i=1}^{k} c_i \mathbb{E}\left[ u_t((\boldsymbol{w}_i^*)^T \boldsymbol{x}) \cdot h_l\left(\boldsymbol{z}^T \boldsymbol{x}\right) \right] \\
&= |\hat{u}_l| \sum_{i=1}^{k} c_i ((\boldsymbol{w}_i^*)^T \boldsymbol{z})^l.
\end{aligned}
$$

It is easy to see that for $\boldsymbol{z} \in S^{n-1}$, the above is maximized at exactly one of the $\boldsymbol{w}_i$'s (up to sign flip for even $l$) for $l \geq 3$ as long as $u_l \neq 0$. Thus, each $\boldsymbol{w}_i$ is a local minima of the above problem.

Let $L(\boldsymbol{z}) = -\sum_{i=1}^{k} c_i z_i^l$. For constraint $||\boldsymbol{z}||^2 = 1$, we have the following optimality conditions (see Nocedal & Wright (2006) for more details).

First order:

$$\nabla L(\boldsymbol{z}) - \frac{\boldsymbol{z}^T \nabla L(\boldsymbol{z})}{||\boldsymbol{z}||^2} \boldsymbol{z} = 0 \text{ and } ||\boldsymbol{z}||^2 = 1.$$

This applied to our function gives us that for $\lambda = -\frac{\sum_i c_i z_i^l}{||\boldsymbol{z}||^2}$ ($\lambda < 0$),

$$-l c_i z_i^{l-1} - 2\lambda z_i = 0$$

The above implies that either $z_i = 0$ or $z_i^{l-2} = -\frac{\lambda}{lc_i}$ with $||\boldsymbol{z}||_2 = 1$. For this to hold $\boldsymbol{z}$ is such that for some set $S \subseteq [n]$, $|S| > 1$, only $i \in S$ have $z_i \neq 0$ and $\sum_{i \in S} z_i^2 = 1$. This implies that for all $i \in S$, $z_i^{l-2} = -\frac{2\lambda}{lc_i}$.

Second order:

$$\text{For all } \boldsymbol{w} \neq 0 \text{ such that } \boldsymbol{w}^T \boldsymbol{z} = 0, \boldsymbol{w}^T (\nabla^2 L(\boldsymbol{z}) - 2\lambda \mathbf{I}) \boldsymbol{w} \geq 0.$$

For our function, we have:

$$\nabla^2 L(\boldsymbol{z}) = -l(l-1)\text{diag}(\boldsymbol{c} \cdot \boldsymbol{z})^{l-2}$$

$$\implies (\nabla^2 L(\boldsymbol{z}))_{ij} = \begin{cases} 2(l-1)\lambda & \text{if } i = j \text{ and } i \in S \\ 0 & \text{otherwise.} \end{cases}$$

The last follows from using the first order condition. For the second order condition to be satisfied we will show that $|S| = 1$. Suppose $|S| > 2$, then choosing $\boldsymbol{w}$ such that $w_i = 0$ for $i \notin S$ and such that $\boldsymbol{w}^T \boldsymbol{z} = 0$ (it is possible to choose such a value since $|S| > 2$), we get $\boldsymbol{w}^T (\nabla^2 L(\boldsymbol{z}) - 2\lambda \mathbf{I}) \boldsymbol{w} = 2(l-2)\lambda ||\boldsymbol{w}||^2$ which is negative since $\lambda < 0$, thus these cannot be global minima. However, for $|S| = 1$, we cannot have such a $\boldsymbol{w}$, since to satisfy $\boldsymbol{w}^T \boldsymbol{z} = 0$, we need $w_i = 0$ for all $i \in S$, this gives us $\boldsymbol{w}^T (\nabla^2 L(\boldsymbol{z}) - 2\lambda \mathbf{I}) \boldsymbol{w} = -2\lambda ||\boldsymbol{w}||^2$ which is always positive. Thus $\boldsymbol{z} = \pm \boldsymbol{e}_i$ are the only local minimas of this problem.

### B.2 IMPORTANT RESULTS FROM GE ET AL. (2017)

**Lemma 13** (Ge et al. (2017)). *If $\boldsymbol{z}$ is an $(\epsilon, \tau)$-local minima of $F(\boldsymbol{z}) = -\sum_i \alpha_i z_i^4 + \lambda(\sum_i z_i^2 - 1)^2$ for $\epsilon \leq \sqrt{\tau^3/\alpha_{\min}}$ where $\alpha_{\min} = \min_i \alpha_i$, then*

- *(Lemma 5.2) $|\boldsymbol{z}|_{2nd} \leq \sqrt{\frac{\tau}{\alpha_{\min}}}$ where $|\boldsymbol{z}|_{2nd}$ denotes the magnitude of the second largest entry in terms of magnitude of $\boldsymbol{z}$.*

- *(Derived from Proposition 5.7) $\boldsymbol{z}_{\max} = \pm 1 \pm O(d\tau/\alpha_{\min}) \pm O(\epsilon/\lambda)$ where $|\boldsymbol{z}|_{\max}$ is the value of the largest entry in terms of magnitude of $\boldsymbol{z}$.*

### B.3 OMITTED PROOFS FOR ONE-BY-ONE RECOVERY

*Proof of Lemma 1.* Let $\mathbf{O} \in \mathbb{R}^{d \times d}$ be the orthonormal basis (row-wise) of the subspace spanned by $\boldsymbol{w}_i^*$ for all $i \in [d]$ generated using Gram-schmidt (with the procedure done in order with elements of $|S|$ first). Now let $\mathbf{O}_S \in \mathbb{R}^{|S| \times d}$ be the matrix corresponding to the first $S$ rows and let $\mathbf{O}_{\bar{S}}^{\perp} \in \mathbb{R}^{(d-|S|) \times n}$ be that corresponding to the remaining rows. Note that $\mathbf{O}\mathbf{W}^*$ ($\mathbf{W}^*$ also has the same ordering) is an upper triangular matrix under this construction.

$$\mathbb{E}\left[\prod_{j \in S} u_t((\boldsymbol{w}_j^*)^T \boldsymbol{x})\right]$$

$$= \frac{1}{(2\pi)^{n/2}} \int_{\boldsymbol{x}} \prod_{i \in S} u_t(\boldsymbol{x}^T \boldsymbol{w}_i^*) e^{-\frac{||\boldsymbol{x}||^2}{2}} d\boldsymbol{x}$$

$$= \frac{1}{(2\pi)^{n/2}} \int_{\boldsymbol{x}} \prod_{i \in S} u_t((\mathbf{O}_S \boldsymbol{w}_i^*)^T \mathbf{O}_S x) e^{-\frac{||\mathbf{O}_S \boldsymbol{x}||^2 + ||\mathbf{O}_{\bar{S}}^{\perp} \boldsymbol{x}||^2}{2}} d\boldsymbol{x}$$

$$= \left(\frac{1}{(2\pi)^{\frac{|S|}{2}}} \int_{\boldsymbol{x}' \in \mathbb{R}^{|S|}} \prod_{i \in S} u_t((\mathbf{O}_S \boldsymbol{w}_i^*)^T \boldsymbol{x}') e^{-\frac{||\boldsymbol{x}'||^2}{2}} d\boldsymbol{x}'\right) \left(\frac{1}{(2\pi)^{\frac{d-|S|}{2}}} \int_{\boldsymbol{x}' \in \mathbb{R}^{d-|S|}} e^{-\frac{||\boldsymbol{x}'||^2}{2}} d\boldsymbol{x}'\right)$$

$$= \frac{1}{(2\pi)^{\frac{|S|}{2}}} \int_{\boldsymbol{x}' \in \mathbb{R}^{|S|}} \prod_{i \in S} u_t((\mathbf{O}_S \boldsymbol{w}_i^*)^T \boldsymbol{x}') e^{-\frac{||\boldsymbol{x}'||^2}{2}} d\boldsymbol{x}'$$

$$= \frac{|\det(\mathbf{O}_S \mathbf{W}_S^*)|^{-1}}{(2\pi)^{\frac{|S|}{2}}} \int_{\boldsymbol{b} \in \mathbb{R}^{|S|}} \prod_{i \in S} u_t(b_i) e^{-\frac{||(\mathbf{O}_S \mathbf{W}_S^*)^{-T} \boldsymbol{b}||^2}{2}} d\boldsymbol{b}$$

Now observe that $\mathbf{O}_S \mathbf{W}_S^*$ is also an upper triangular matrix since it is a principal sub-matrix of $\mathbf{O}\mathbf{W}^*$. Thus using Fact 6 and 7, we get the last equality. Also, the single non-zero entry row has non-zero entry being 1 ($||\boldsymbol{w}_i^*|| = 1$ for all $i$). This gives us that the inverse will also have the single non-zero entry row has non-zero entry being 1. WLOG assume index 1 corresponds to this row. Thus we can split this as following

$$\mathbb{E}\left[\prod_{j \in S} u_t((\boldsymbol{w}_j^*)^T \boldsymbol{x})\right]$$

$$\leq |\det(\mathbf{O}_S \mathbf{W}_S^*)|^{-1} \left(\frac{1}{\sqrt{2\pi}} \int_{b_1} u_t(b_1) e^{-\frac{b_1^2}{2}} db_1\right) \left(\prod_{i \in S \setminus \{1\}} \frac{1}{\sqrt{2\pi}} \int_{b_i} u_t(b_i) e^{-\frac{b_i^2}{2||\mathbf{O}_S \mathbf{W}_S^*||^2}} db_i\right)$$

$$\leq |\det(\mathbf{O}_S \mathbf{W}_S^*)|^{-1} \left(\frac{1}{\sqrt{2\pi}} \int_{b_1} u_t(b_1) e^{-\frac{b_1^2}{2}} db_1\right) \left(\prod_{i \in S \setminus \{1\}} \frac{1}{\sqrt{2\pi}} \int_{b_i} u_t(b_i) e^{-\frac{b_i^2}{||\mathbf{W}^*||^2}} db_i\right)$$

$$\leq \rho(t, 1) \left(\kappa(\mathbf{W}^*) \rho(t, ||\mathbf{W}^*||)\right)^{|S|-1}$$

$\square$

*Proof of Claim 1.* Consider the SVD of matrix $\mathbf{M} = \mathbf{U}\mathbf{D}\mathbf{U}^T$. Let $\mathbf{W} = \mathbf{U}\mathbf{D}^{-1/2}$ and $\boldsymbol{y}_i = \sqrt{c_i} \mathbf{W}^T \boldsymbol{w}_i^*$ for all $i$. It is easy to see that $\boldsymbol{y}_i$ are orthogonal. Let $F(\boldsymbol{z}) = G(\mathbf{W}\boldsymbol{z})$:

$$F(\boldsymbol{z}) = |\hat{u}_4| \sum_i c_i (\boldsymbol{z}^T \mathbf{W}^T \boldsymbol{w}_i^*)^4 - \lambda \hat{u}_2^2 \left(\sum_i c_i (\boldsymbol{z}^T \mathbf{W}^T \boldsymbol{w}_i^*)^2 - 1\right)^2$$

$$= |\hat{u}_4| \sum_i \frac{1}{c_i} (\boldsymbol{z}^T \boldsymbol{y}_i)^4 - \lambda \hat{u}_2^2 \left(\sum_i (\boldsymbol{z}^T \boldsymbol{y}_i)^2 - 1\right)^2.$$

Since $\boldsymbol{y}_i$ are orthogonal, for means of analysis, we can assume that $\boldsymbol{y}_i = \boldsymbol{e}_i$, thus the formulation reduces to $\max_{\boldsymbol{z}} |\hat{u}_4| \sum_i \frac{1}{c_i} (z_i)^4 - \lambda' \left(||\boldsymbol{z}||^2 - 1\right)^2$ up to scaling of $\lambda' = \lambda \hat{u}_2^2$. Note that this is of the form in Lemma 13 hence using that we can show that the approximate local minimas of $F(\boldsymbol{z})$ are close to $\boldsymbol{y}_i$ and thus the local maximas of $G(\boldsymbol{z})$ are close to $\mathbf{W}\boldsymbol{y}_i = \sqrt{c_i} \mathbf{W}\mathbf{W}^T \boldsymbol{w}_i^* = \sqrt{c_i} \mathbf{M}^{-1} \boldsymbol{w}_i^*$ due to the linear transformation. This can alternately be viewed as the columns of $(\mathbf{T}\mathbf{W}^*)^{-1}$ since $\mathbf{T}\mathbf{W}^* \mathbf{M}^{-1} (\mathbf{T}\mathbf{W}^*)^T = \mathbf{I}$. $\square$

*Proof of Theorem 4.* Let $\mathbf{Z}$ be an $(\epsilon, \tau)$-local minimum of $A$, then we have $||\nabla A(\mathbf{Z})|| \leq \epsilon$ and $\lambda_{\min}(\nabla^2 A(\mathbf{Z})) \geq -\tau$. Observe that

$$||\nabla B(\mathbf{Z})|| = ||\nabla(A + (B - A)(\mathbf{Z})|| \leq ||\nabla A(\mathbf{Z})|| + ||\nabla(B - A)(\mathbf{Z})|| \leq \epsilon + \rho.$$

Also observe that

$$\lambda_{\min}(\nabla^2 B(\mathbf{Z})) = \lambda_{\min}(\nabla^2(A + (B - A))(\mathbf{Z}))$$
$$\geq \lambda_{\min}(\nabla^2 A(\mathbf{Z})) + \lambda_{\min}(\nabla^2(B - A)(\mathbf{Z}))$$
$$\geq -\tau - ||\nabla^2(B - A)(\mathbf{Z})|| \geq -\tau - \gamma$$

Here we use $|\lambda_{\min}(\mathbf{M})| \leq ||\mathbf{M}||$ for any symmetric matrix. To prove this, we have $||\mathbf{M}|| = \max_{\boldsymbol{x} \in \mathbb{S}^{n-1}} ||\mathbf{M}\boldsymbol{x}||$. We have $\boldsymbol{x} = \sum_i x_i \boldsymbol{v}_i$ where $\boldsymbol{v}_i$ are the eigenvectors. Thus we have $\mathbf{M}\boldsymbol{x} = \sum_i x_i \lambda_i(\mathbf{M}) \boldsymbol{v}_i$ and $\sum_i x_i^2 = 1$. Which gives us that $||\mathbf{M}|| = \sqrt{\sum_i x_i^2 \lambda_i^2(\mathbf{M})} \geq |\lambda_{\min}(\mathbf{M})|$. $\square$

*Proof of Lemma 2.* Expanding $f$, we have

$$\mathbb{E}[|\Delta(\boldsymbol{x})|] = \mathbb{E}\left[\left|\sum_{S\subseteq[d]:|S|>1} c_S \prod_{j\in S} u_t((\boldsymbol{w}_j^*)^T\boldsymbol{x})\right|\right]$$

$$\leq \sum_{S\subseteq[d]:|S|>1} |c_S|\mathbb{E}\left[\prod_{j\in S} u_t((\boldsymbol{w}_j^*)^T\boldsymbol{x})\right]$$

using Lemma 1
$$\leq C \sum_{S\subseteq[d]:|S|>1} \rho(t,1)\left(\frac{1}{\sigma_{\min}(\mathbf{W}^*)}\rho(t,||\mathbf{W}^*||)\right)^{|S|-1}$$

$$= C\sum_{i=1}^{d} \binom{d}{i}\rho(t,1)\left(\frac{1}{\sigma_{\min}(\mathbf{W}^*)}\rho(t,||\mathbf{W}^*||)\right)^{i-1}$$

using $\binom{d}{i} \leq d^i$
$$\leq C\sum_{i=1}^{d} d\rho(t,1)\left(\frac{d}{\sigma_{\min}(\mathbf{W}^*)}\rho(t,||\mathbf{W}^*||)\right)^{i-1}$$

using assumption on $t$
$$\leq Cd^2\rho(t,1)\left(\frac{d}{\sigma_{\min}(\mathbf{W}^*)}\rho(t,||\mathbf{W}^*||)\right)$$

$\square$

**Lemma 14.** *For any function $L$ such that $||L(\boldsymbol{z},\boldsymbol{x})|| \leq C(\boldsymbol{z})||\boldsymbol{x}||^{O(1)}$ where $C$ is a function that is not dependent on $\boldsymbol{x}$, we have $||\mathbb{E}[\Delta(\boldsymbol{x})L(\boldsymbol{x})]|| \leq C(\boldsymbol{z})d^{-(1+p)\eta+3}O(\log d)$.*

*Proof.* We have

$$||\mathbb{E}[\Delta(\boldsymbol{x})L(\boldsymbol{x})]||$$
$$\leq \mathbb{E}[|\Delta(\boldsymbol{x})||L(\boldsymbol{x})|]$$
$$\leq \mathbb{E}[|\Delta(\boldsymbol{x})C(\boldsymbol{z})||\boldsymbol{x}||^{O(1)}]$$
$$= C(\boldsymbol{z})\left(\mathbb{E}[|\Delta(\boldsymbol{x})|\ ||\boldsymbol{x}||^{O(1)}|\ ||\boldsymbol{x}|| \geq c]Pr[||\boldsymbol{x}|| \geq c]\right.$$
$$\left. + \mathbb{E}[|\Delta(\boldsymbol{x})|\ ||\boldsymbol{x}||^{O(1)}|\ ||\boldsymbol{x}|| < c]Pr[||\boldsymbol{x}|| < c]\right)$$
$$\leq C(\boldsymbol{z})(\mathbb{E}[||\boldsymbol{x}||^{O(1)}|||\boldsymbol{x}|| \geq c]Pr[||\boldsymbol{x}|| \geq c] + c\mathbb{E}[|\Delta(\boldsymbol{x})|])$$
$$= C(\boldsymbol{z})(c^{O(1)}e^{-\frac{c^2}{2}} + c^{O(1)}\mathbb{E}[|\Delta(\boldsymbol{x})|]).$$

Now using Lemma 2 to bound $\mathbb{E}[|\Delta(\boldsymbol{x})|]$, for $c = \Theta(\sqrt{\eta\log d})$ we get the required result. $\square$

**Lemma 15.** *For $||\boldsymbol{z}|| = \Omega(1)$ and $\lambda = \Theta(|\hat{u}_4|/\hat{u}_2^2) \approx d^\eta$, $||\nabla G(\boldsymbol{z})|| \geq \Omega(1)d^{-\eta}$.*

*Proof.* Let $K = \kappa(\mathbf{W}^*)$ which by assumption is $\theta(1)$. We will argue that local minima of $G$ cannot have $\boldsymbol{z}$ with large norm. First lets argue this for $G_{\text{lin}}(\boldsymbol{z})$. We know that $G_{\text{lin}}(\boldsymbol{z}) = -\alpha\sum(\boldsymbol{z}^T\boldsymbol{w}_i^*)^4 + \lambda\beta^2((\sum(\boldsymbol{z}^T\boldsymbol{w}_i^*)^2)-1)^2$ where $\alpha = |\hat{u}_4|$ and $\beta = \hat{u}_2$. We will argue that $\boldsymbol{z}^T\nabla G_{\text{lin}}(\boldsymbol{z})$ is large if $\boldsymbol{z}$ is large.

$$\boldsymbol{z}^T\nabla G_{\text{lin}}(\boldsymbol{z}) = -4\alpha\sum(\boldsymbol{z}^T\boldsymbol{w}_i^*)^3(\boldsymbol{z}^T\boldsymbol{w}_i^*) + 2\lambda\beta^2\left(\sum(\boldsymbol{z}^T\boldsymbol{w}_i^*)^2 - 1\right)\left(\sum 2(\boldsymbol{z}^T\boldsymbol{w}_i^*)(\boldsymbol{z}^T\boldsymbol{w}_i^*)\right)$$
$$= -4\alpha\sum(\boldsymbol{z}^T\boldsymbol{w}_i^*)^4 + 4\lambda\beta^2\left(\sum(\boldsymbol{z}^T\boldsymbol{w}_i^*)^2 - 1\right)\left(\sum(\boldsymbol{z}^T\boldsymbol{w}_i^*)^2\right)$$

Let $\boldsymbol{y} = \mathbf{W}^*\boldsymbol{z}$ then $K||\boldsymbol{z}|| \geq ||\boldsymbol{y}|| \geq ||\boldsymbol{z}||/K$ since $K$ is the condition number of $\mathbf{W}^*$. Then this implies

$$\boldsymbol{z}^T\nabla G_{\text{lin}}(\boldsymbol{z}) = -4\alpha\sum y_i^4 + 4\lambda\beta^2(||\boldsymbol{y}||^2 - 1)||\boldsymbol{y}||^2$$
$$= 4||\boldsymbol{y}||^2((-\alpha + \lambda\beta^2)||\boldsymbol{y}||^2 + \lambda\beta^2)$$
$$\geq ||\boldsymbol{y}||^4(-\alpha + \lambda\beta^2) \geq \Omega(1)d^{-\eta}||\boldsymbol{y}||^4$$

Since $||\boldsymbol{y}|| \geq ||\boldsymbol{z}||/K = \Omega(1)$ by assumptions on $\lambda, \boldsymbol{z}$ we have $\boldsymbol{z}^T \nabla G_{\text{lin}}(\boldsymbol{z}) \geq \Omega(\lambda \beta^2 ||\boldsymbol{y}||^4) = \Omega(1) d^{-\eta} ||\boldsymbol{z}||^4$. This implies $||\nabla G_{\text{lin}}(\boldsymbol{z})|| = \Omega(1) d^{-\eta} ||\boldsymbol{z}||^3$.

Now we need to argue for $G$.

$G(\boldsymbol{z}) - G_{\text{lin}}(\boldsymbol{z})$

$= -\text{sgn}(\hat{u}_4) \mathbb{E}[(f_{\text{lin}}(\boldsymbol{x}) + \Delta(\boldsymbol{x})) H_4(\boldsymbol{z}^T \boldsymbol{x})] + \lambda (\mathbb{E}[(f_{\text{lin}}(\boldsymbol{x}) + \Delta(\boldsymbol{x})) H_2(\boldsymbol{z}^T \boldsymbol{x})] - \beta)^2$

$\quad + \text{sgn}(\hat{u}_4) \mathbb{E}[(f_{\text{lin}}(\boldsymbol{x})) H_4(\boldsymbol{z}^T \boldsymbol{x})] - \lambda \mathbb{E}[(f_{\text{lin}}(\boldsymbol{x})) H_2(\boldsymbol{z}^T \boldsymbol{x})] - \beta]^2$

$= -\text{sgn}(\hat{u}_4) \mathbb{E}[\Delta(\boldsymbol{x}) H_4(\boldsymbol{z}^T \boldsymbol{x})] + \lambda \mathbb{E}[\Delta(\boldsymbol{x}) H_2(\boldsymbol{z}^T \boldsymbol{x})]^2 + 2\lambda \mathbb{E}[\Delta(\boldsymbol{x}) H_2(\boldsymbol{z}^T \boldsymbol{x})] \mathbb{E}[f_{\text{lin}}(\boldsymbol{x}) H_2(\boldsymbol{z}^T \boldsymbol{x}) - \beta]$

$= -\text{sgn}(\hat{u}_4) ||\boldsymbol{z}||^4 \mathbb{E}[\Delta(\boldsymbol{x}) h_4(\boldsymbol{z}^T \boldsymbol{x}/||\boldsymbol{z}||)] + \lambda ||\boldsymbol{z}||^4 \mathbb{E}[\Delta(\boldsymbol{x}) h_2(\boldsymbol{z}^T \boldsymbol{x}/||\boldsymbol{z}||)]^2$

$\quad + 2\lambda ||\boldsymbol{z}||^4 \mathbb{E}[\Delta(\boldsymbol{x}) h_2(\boldsymbol{z}^T \boldsymbol{x}/||\boldsymbol{z}||)] \mathbb{E}[f_{\text{lin}}(\boldsymbol{x}) h_2(\boldsymbol{z}^T \boldsymbol{x}/||\boldsymbol{z}||)] - 2\lambda \beta ||\boldsymbol{z}||^2 \mathbb{E}[\Delta(\boldsymbol{x}) h_2(\boldsymbol{z}^T \boldsymbol{x}/||\boldsymbol{z}||)]$

Now $h_4(\boldsymbol{z}^T \boldsymbol{x}/||\boldsymbol{z}||)$ doesn't have a gradient in the direction of $\boldsymbol{z}$ so $\boldsymbol{z}^T \nabla h_4(\boldsymbol{z}^T \boldsymbol{x}/||\boldsymbol{z}||) = 0$. Similarly $\boldsymbol{z}^T \nabla h_2(\boldsymbol{z}^T \boldsymbol{x}/||\boldsymbol{z}||) = 0$. So

$\boldsymbol{z}^T \nabla (G(\boldsymbol{z}) - G_{\text{lin}}(\boldsymbol{z}))$

$= -4\text{sgn}(\hat{u}_4) ||\boldsymbol{z}||^4 \mathbb{E}[\Delta(\boldsymbol{x}) h_4(\boldsymbol{z}^T \boldsymbol{x}/||\boldsymbol{z}||)] + 4\lambda ||\boldsymbol{z}||^4 (\mathbb{E}[\Delta(\boldsymbol{x}) h_2(\boldsymbol{z}^T \boldsymbol{x}/||\boldsymbol{z}||)])^2$

$\quad + 8\lambda ||\boldsymbol{z}||^4 \mathbb{E}[\Delta(\boldsymbol{x}) h_2(\boldsymbol{z}^T \boldsymbol{x}/||\boldsymbol{z}||)] \mathbb{E}[f_{\text{lin}}(\boldsymbol{x}) h_2(\boldsymbol{z}^T \boldsymbol{x}/||\boldsymbol{z}||)] - 4\lambda \beta ||\boldsymbol{z}||^2 \mathbb{E}[\Delta(\boldsymbol{x}) h_2(\boldsymbol{z}^T \boldsymbol{x}/||\boldsymbol{z}||)]$

We know that $\mathbb{E}[f_{\text{lin}}(\boldsymbol{x}) h_2(\boldsymbol{z}^T \boldsymbol{x}/||\boldsymbol{z}||)]$ has a factor of $\beta$ giving us using Lemma 14:

$$|\boldsymbol{z}^T \nabla (G(\boldsymbol{z}) - G_{\text{lin}}(\boldsymbol{z}))| \leq O(\log d) d^{-(1+p)\eta + 3} ||\boldsymbol{z}||^4.$$

So $\boldsymbol{z}^T \nabla G(z)$ is also $\Omega(||\boldsymbol{z}||^4)$. so $||\nabla G(\boldsymbol{z})|| \geq \Omega(1) d^{-\eta}$ $\qquad \square$

*Proof of Claim 2.* We have $G - G_{\text{lin}}$ as follows,

$G(\boldsymbol{z}) - G_{\text{lin}}(\boldsymbol{z})$

$= -\text{sgn}(\hat{u}_4) \mathbb{E}[(f_{\text{lin}}(\boldsymbol{x}) + \Delta(\boldsymbol{x})) H_4(\boldsymbol{z}^T \boldsymbol{x})] + \lambda (\mathbb{E}[(f_{\text{lin}}(\boldsymbol{x}) + \Delta(\boldsymbol{x})) H_2(\boldsymbol{z}^T \boldsymbol{x})] - \hat{u}_2)^2$

$\quad + \text{sgn}(\hat{u}_4) \mathbb{E}[(f_{\text{lin}}(\boldsymbol{x})) H_4(\boldsymbol{z}^T \boldsymbol{x})] - \lambda (\mathbb{E}[(f_{\text{lin}}(\boldsymbol{x})) H_2(\boldsymbol{z}^T \boldsymbol{x})] - \hat{u}_2)^2$

$= -\text{sgn}(\hat{u}_4) \mathbb{E}[\Delta(\boldsymbol{x}) H_4(\boldsymbol{z}^T \boldsymbol{x})] + \lambda (\mathbb{E}[\Delta(\boldsymbol{x}) H_2(\boldsymbol{z}^T \boldsymbol{x})])^2$

$\quad + 2\lambda \mathbb{E}[\Delta(\boldsymbol{x}) H_2(\boldsymbol{z}^T \boldsymbol{x})] \mathbb{E}[f_{\text{lin}}(\boldsymbol{x}) H_2(\boldsymbol{z}^T \boldsymbol{x}) - \hat{u}_2]$

Thus we have,

$\nabla (G(\boldsymbol{z}) - G_{\text{lin}}(\boldsymbol{z}))$

$= -\text{sgn}(\hat{u}_4) \mathbb{E}[\Delta(\boldsymbol{x}) \nabla H_4(\boldsymbol{z}^T \boldsymbol{x})] + 2\lambda \mathbb{E}[\Delta(\boldsymbol{x}) H_2(\boldsymbol{z}^T \boldsymbol{x})] \mathbb{E}[\Delta(\boldsymbol{x}) \nabla H_2(\boldsymbol{z}^T \boldsymbol{x})]$

$\quad + 2\lambda \mathbb{E}[f_{\text{lin}}(\boldsymbol{x}) H_2(\boldsymbol{z}^T \boldsymbol{x}) - \hat{u}_2] \mathbb{E}[\Delta(\boldsymbol{x}) \nabla H_2(\boldsymbol{z}^T \boldsymbol{x})]$

$\quad + 2\lambda \mathbb{E}[\Delta(\boldsymbol{x}) H_2(\boldsymbol{z}^T \boldsymbol{x})] \mathbb{E}[f_{\text{lin}}(\boldsymbol{x}) \nabla H_2(\boldsymbol{z}^T \boldsymbol{x})]$

Observe that $H_2$ and $H_4$ are degree 2 and 4 (respectively) polynomials thus norm of gradient and hessian of the same can be bounded by at most $O(||\boldsymbol{z}|| ||\boldsymbol{x}||^4)$. Using Lemma 14 we can bound each term by roughly $O(\log d) d^{-(1+p)\eta + 3} ||\boldsymbol{z}||^4$. Note that $\lambda$ being large does not hurt as it is scaled appropriately in each term. Subsequently, using Lemma 15, we can show that $||\boldsymbol{z}||$ is bounded by a constant since $||G(\boldsymbol{z})|| \leq d^{-2\eta}$. Similar analysis holds for the hessian too.

Now applying Theorem 4 gives us that $\boldsymbol{z}$ is an $(O(\log d) d^{-(1+p)\eta + 3}, O(\log d) d^{-(1+p)\eta + 3})$-approximate local minima of $G_{\text{lin}}$. This implies that it is also an $(\epsilon' := C \log(d) d^{-(1+2p)\eta + 3}, \tau' := C \log(d) d^{-(1+2p/3)\eta + 3})$-approximate local minima of $G_{\text{lin}}$ for large enough $C > 0$ by increasing $\tau$. Observe that $\sqrt{\tau'^3/|\hat{u}_4|} = C^{3/2} \log^{3/2}(d) d^{-(3/2+p)\eta + 9/2}/d^{-\eta/2} = C^{3/2} \log^{3/2}(d) d^{-(1+p)\eta + 9/2} \geq \epsilon'$. Now using Claim 1, we get the required result. $\qquad \square$

## B.4 SIMULTANEOUS RECOVERY

Ge et al. (2017) also showed simultaneous recovery by minimizing the following loss function $G_{\text{lin}}$ defined below has a well-behaved landscape.

$$G_{\text{lin}}(\mathbf{W}) = \mathbb{E}\left[ f_{\text{lin}}(\boldsymbol{x}) \sum_{j,k \in [d], j \neq k} \psi(\boldsymbol{w}_j, \boldsymbol{w}_k, \boldsymbol{x}) \right] - \gamma \mathbb{E}\left[ f_{\text{lin}}(\boldsymbol{x}) \sum_{j \in [d]} H_4(\boldsymbol{w}_j^T \boldsymbol{x}) \right] \qquad (1)$$

$$+ \lambda \sum_i \left( \mathbb{E}\left[ f_{\text{lin}}(\boldsymbol{x}) H_2(\boldsymbol{w}_i^T \boldsymbol{x}) \right] - \hat{u}_2 \right)^2 \tag{2}$$

where $\psi(v, w, \boldsymbol{x}) = H_2(\boldsymbol{v}^T \boldsymbol{x}) H_2(\boldsymbol{w}^T \boldsymbol{x}) + 2(\boldsymbol{v}^T \boldsymbol{w})^2 + 4(\boldsymbol{v}^T \boldsymbol{x})(\boldsymbol{w}^T \boldsymbol{x}) \boldsymbol{v}^T \boldsymbol{w}$.

They gave the following result.

**Theorem 10** (Ge et al. (2017)). *Let $c$ be a sufficiently small universal constant (e.g. $c = 0.01$ suffices), and suppose the activation function $u$ satisfies $\hat{u}_4 \neq 0$. Assume $\gamma \leq c, \lambda \geq \Omega(|\hat{u}_4|/\hat{u}_2^2)$, and $\mathbf{W}^*$ be the true weight matrix. The function $G_{\text{lin}}$ satisfies the following:*

1. *Any saddle point $\mathbf{W}$ has a strictly negative curvature in the sense that $\lambda_{\min}(\nabla^2 G_{\text{lin}}(\mathbf{W})) \geq -\tau_0$ where $\tau_0 = c \min\{\gamma|\hat{u}_4|/d, \lambda \hat{u}_2^2\}$.*

2. *Suppose $\mathbf{W}$ is an $(\epsilon, \tau_0)$-approximate local minimum, then $\mathbf{W}$ can be written as $\mathbf{W}^{-T} = \mathbf{PDW}^* + \mathbf{E}$ where $\mathbf{D}$ is a diagonal matrix with $D_{ii} \in \{\pm 1 \pm O(\gamma|\hat{u}_4|/\lambda\hat{u}_2^2) \pm O(\epsilon/\lambda)\}$, $\mathbf{P}$ is a permutation matrix, and the error term $||\mathbf{E}|| \leq O(\epsilon d/\hat{u}_4)$.*

We show that this minimization is robust. Let us consider the corresponding function $G$ to $G_{\text{lin}}$ with the additional non-linear terms as follows:

$$G(\mathbf{W}) = \mathbb{E}\left[ f(\boldsymbol{x}) \sum_{j,k \in [d], j \neq k} \psi(\boldsymbol{w}_j, \boldsymbol{w}_d, \boldsymbol{x}) \right] - \gamma \mathbb{E}\left[ f(\boldsymbol{x}) \sum_{j \in [d]} H_4(\boldsymbol{w}_j, \boldsymbol{x}) \right]$$
$$+ \lambda \sum_i \left( \mathbb{E}\left[ f(\boldsymbol{x}) H_2(\boldsymbol{w}_i, \boldsymbol{x}) \right] - \hat{u}_2 \right)^2$$

Now we can show that $G$ and $G_{\text{lin}}$ are close as in the one-by-one case.

$R(\mathbf{W}) := G(\mathbf{W}) - G_{\text{lin}}(\mathbf{W})$

$= \mathbb{E}\left[\Delta(\boldsymbol{x}) A(\mathbf{W}, \boldsymbol{x})\right] - \gamma \mathbb{E}\left[\Delta(\boldsymbol{x}) B(\mathbf{W}, \boldsymbol{x})\right] + \lambda \left( \mathbb{E}\left[f(\boldsymbol{x}) C(\mathbf{W}, \boldsymbol{x})\right]^2 - \mathbb{E}\left[f_{\text{lin}}(\boldsymbol{x}) C(\mathbf{W}, \boldsymbol{x})\right]^2 \right)$

$= \mathbb{E}\left[\Delta(\boldsymbol{x}) A(\mathbf{W}, \boldsymbol{x})\right] - \gamma \mathbb{E}\left[\Delta(\boldsymbol{x}) B(\mathbf{W}, \boldsymbol{x})\right] + \lambda \mathbb{E}\left[(\Delta(\boldsymbol{x}) C(\mathbf{W}, \boldsymbol{x})(f(\boldsymbol{x}') + f_{\text{lin}}(\boldsymbol{x}')) C(\mathbf{W}, \boldsymbol{x}')\right]$

$= \mathbb{E}\left[\Delta(\boldsymbol{x}) A(\mathbf{W}, \boldsymbol{x})\right] - \gamma \mathbb{E}\left[\Delta(\boldsymbol{x}) B(\mathbf{W}, \boldsymbol{x})\right] + \lambda \mathbb{E}\left[(\Delta(\boldsymbol{x}) D(\mathbf{W}, \boldsymbol{x})\right]$

$= \mathbb{E}\left[\Delta(\boldsymbol{x})(A(\mathbf{W}, \boldsymbol{x}) - \gamma B(\mathbf{W}, \boldsymbol{x}) + \lambda D(\mathbf{W}, \boldsymbol{x}))\right]$

$= \mathbb{E}\left[\Delta(\boldsymbol{x}) L(\mathbf{W}, \boldsymbol{x})\right]$

where $A(\mathbf{W}, \boldsymbol{x}) = \sum_{j,k \in [d], j \neq k} \psi(\boldsymbol{w}_j, \boldsymbol{w}_d, \boldsymbol{x})$, $B(\mathbf{W}, \boldsymbol{x}) = \sum_{j \in [d]} H_4(\boldsymbol{w}_j, \boldsymbol{x})$, $C(\mathbf{W}, \boldsymbol{x}) = \sum_i H_2(\boldsymbol{w}_i, \boldsymbol{x})$, $D(\mathbf{W}, \boldsymbol{x}) = C(\mathbf{W}, \boldsymbol{x}) \mathbb{E}[(f(\boldsymbol{x}') + f_{\text{lin}}(\boldsymbol{x}')) C(\mathbf{W}, \boldsymbol{x}')]$ and $L(\mathbf{W}, \boldsymbol{x}) = A(\mathbf{W}, \boldsymbol{x}) - \gamma B(\mathbf{W}, \boldsymbol{x}) + \lambda D(\mathbf{W}, \boldsymbol{x})$.

Using similar analysis as the one-by-one case, we can show the required closeness. It is easy to see that $||\nabla L||$ and $||\nabla^2 L||$ will be bounded above by a constant degree polynomial in $O(\log d) d^{-(1+p)\eta+3} \max ||\boldsymbol{w}_i||^4$. No row can have large weight as if any row is large, then looking at the gradient for that row, it reduces to the one-by-one case, and there it can not be larger than a constant. Thus we have the same closeness as in the one-by-one case. Combining this with Theorem 10 and 4, we have the following theorem:

**Theorem 11.** *Let $c$ be a sufficiently small universal constant (e.g. $c = 0.01$ suffices), and under Assumptions 1, 2 and 3. Assume $\gamma \leq c, \lambda = \Theta(d^\eta)$, and $\mathbf{W}^*$ be the true weight matrix. The function $G$ satisfies the following*

1. *Any saddle point $\mathbf{W}$ has a strictly negative curvature in the sense that $\lambda_{\min}(\nabla^2 G_{\text{lin}}(\mathbf{W})) \geq -\tau$ where $\tau_0 = O(\log d) d^{-\Omega(1)}$.*

2. *Suppose $\mathbf{W}$ is a $(d^{-\Omega(1)}, d^{-\Omega(1)})$-approximate local minimum, then $\mathbf{W}$ can be written as $\mathbf{W}^{-T} = \mathbf{PDW}^* + \mathbf{E}$ where $\mathbf{D}$ is a diagonal matrix with $D_{ii} \in \{\pm 1 \pm O(\gamma) \pm d^{-\Omega(1)}\}$, $\mathbf{P}$ is a permutation matrix, and the error term $||\mathbf{E}|| \leq O(\log d) d^{-\Omega(1)}$.*

Using standard optimization techniques we can find a local minima.

## B.5 APPROXIMATE TO ARBITRARY CLOSE

**Lemma 16.** *If $u$ is the sign function then $\mathbb{E}[u(\boldsymbol{w}^T\boldsymbol{x})\delta'(\boldsymbol{z}^T\boldsymbol{x})] = c|\cot(\alpha)|$ where $\boldsymbol{w}, \boldsymbol{z}$ are unit vectors and $\alpha$ is the angle between them and $c$ is some constant.*

*Proof.* WLOG we can work the in the plane spanned by $\boldsymbol{z}$ and $\boldsymbol{w}$ and assume that $\boldsymbol{z}$ is the vector $\boldsymbol{i}$ along and $\boldsymbol{w} = \boldsymbol{i}\cos\alpha + \boldsymbol{j}\sin\alpha$. Thus we can replace the vector $\boldsymbol{x}$ by $\boldsymbol{i}x + \boldsymbol{j}y$ where $x, y$ are normally distributed scalars. Also note that $u' = \delta$ (Dirac delta function).

$$\mathbb{E}[u(\boldsymbol{w}^T\boldsymbol{x})\delta'(\boldsymbol{z}^T\boldsymbol{x})] = \mathbb{E}[u(x\cos\alpha + y\sin\alpha)\delta'(x)]$$
$$= \int_y \int_x u(x\cos\alpha + y\sin\alpha)\delta'(x)\phi(x)\phi(y)dxdy$$

Using the fact that $\int_x \delta'(x)h(x)dx = h'(0)$ this becomes

$$= \int_y \phi(y)[(\partial/\partial x)u(x\cos\alpha + y\sin\alpha)\phi(x)]_{x=0}dy$$
$$= \int_y \phi(y)[n(x)u'(x\cos\alpha + y\sin\alpha)\cos\alpha + \phi'(x)u(x\cos\alpha + y\sin\alpha)]_{x=0}dy$$
$$= \int_{y=-\infty}^{\infty} \phi(y)\phi(0)\delta(y\sin\alpha)\cos\alpha\, dy$$

Substituting $s = y\sin\alpha$ this becomes

$$= \int_{s=-\infty/\sin\alpha}^{\infty/\sin\alpha} \phi(s/\sin\alpha)\phi(0)\delta(s)\cos\alpha(1/\sin\alpha)ds$$
$$= \text{sgn}(\sin\alpha)\cot(\alpha)\phi(0)\int_s \phi(s/\sin\alpha)\delta(s)ds$$
$$= |\cot(\alpha)|\phi(0)\phi(0)$$

$\square$

*Proof of Lemma 4.* Let us compute the probability of lying in the $\epsilon$-band for any $t$:

$$Pr[\boldsymbol{x} \in l(\boldsymbol{z}, t, \epsilon)] = Pr[t - \epsilon \le \boldsymbol{z}^T\boldsymbol{x} \le t]$$
$$= \Pr_{g \in N(0, ||\boldsymbol{z}||^2)}[t - \epsilon \le g \le t]$$
$$= \frac{1}{\sqrt{2\pi}||\boldsymbol{z}||}\int_{g=t-\epsilon}^{t} e^{-\frac{g^2}{2||\boldsymbol{z}||^2}}dg \qquad = \frac{\epsilon}{\sqrt{2\pi}||\boldsymbol{z}||}e^{-\frac{\bar{t}^2}{2||\boldsymbol{z}||^2}}$$

where the last equality follows from the mean-value theorem for some $\bar{t} \in [t - \epsilon, t]$.

Next we compute the following:

$$Pr[\boldsymbol{x}^T\boldsymbol{w}_1^* \ge t \text{ and } \boldsymbol{x} \in l(\boldsymbol{z}, t', \epsilon)]$$
$$= \frac{1}{(2\pi)^{\frac{n}{2}}}\int_{\boldsymbol{x}} \text{sgn}(x_1 - t)\mathbb{1}[\boldsymbol{x} \in l(\boldsymbol{z}, t', \epsilon)]e^{-\frac{||\boldsymbol{x}||^2}{2}}d\boldsymbol{x}$$
$$= \frac{1}{(2\pi)^{\frac{1}{2}}}\int_{x_1=t}^{\infty} e^{-\frac{x_1^2}{2}}\left(\frac{1}{(2\pi)^{\frac{n-1}{2}}}\int_{\boldsymbol{x}_{-1}} \mathbb{1}[\boldsymbol{x}_{-1} \in l(\boldsymbol{z}_{-1}, t' - z_1 x_1, \epsilon)]e^{-\frac{||\boldsymbol{x}_{-1}||^2}{2}}d\boldsymbol{x}_{-1}\right)dx_1$$
$$= \frac{1}{(2\pi)^{\frac{1}{2}}}\int_{x_1=t}^{\infty} e^{-\frac{x_1^2}{2}}Pr[\boldsymbol{x}_{-1} \in l(\boldsymbol{z}_{-1}, t - z_1 x_1, \epsilon)]d\boldsymbol{x}_{-1}$$
$$= \frac{\epsilon}{2\pi||\boldsymbol{z}_{-1}||}\int_{g=t'-\epsilon}^{t'}\int_{x_1=t}^{\infty} e^{-\frac{x_1^2}{2}}e^{-\frac{(g-z_1 x_1)^2}{2||\boldsymbol{z}_{-1}||^2}}dx_1 dg$$

$$= \frac{1}{2\pi||\boldsymbol{z}_{-1}||} \int_{g=t'-\epsilon}^{t'} e^{-\frac{g^2}{2||\boldsymbol{z}||^2}} \int_{x_1=t}^{\infty} e^{-\frac{\left(x_1 - \frac{g z_1}{||\boldsymbol{z}||^2}\right)^2}{2\frac{||\boldsymbol{z}_{-1}||^2}{||\boldsymbol{z}||^2}}} \ dx_1 dg$$

$$= \frac{1}{\sqrt{2\pi}||\boldsymbol{z}||} \int_{g=t'-\epsilon}^{t'} e^{-\frac{g^2}{2||\boldsymbol{z}||^2}} \Phi^c \left( \frac{t||\boldsymbol{z}||^2 - g z_1}{||z_{-1}|| \, |||\boldsymbol{z}||} \right) dg$$

$$= \frac{\epsilon}{\sqrt{2\pi}} e^{-\frac{t*^2}{2}} \Phi^c \left( \frac{t - t*\cos(\alpha_1)}{|\sin(\alpha_1)|} \right)$$

where the last equality follows from the mean-value theorem for some $t* \in [t' - \epsilon, t']$. Combining, we get:

$$Pr[\boldsymbol{x}^T \boldsymbol{w}_1* \geq t \text{ and } \boldsymbol{x} \in l(\boldsymbol{z}, t', \epsilon) | \boldsymbol{x} \in l(\boldsymbol{z}, t, \epsilon)]$$

$$= e^{-\frac{t*^2 - \bar{t}^2}{2}} \Phi^c \left( \frac{t - t^* \cos(\alpha_1)}{|\sin(\alpha_1)|} \right) = \Phi^c \left( \frac{t - t^* \cos(\alpha_1)}{|\sin(\alpha_1)|} \right) \pm O(\epsilon) t'$$

for $\epsilon \leq 1/t'$. $\qquad \square$

*Proof of Lemma 5.* Recall that $P$ is monotone with positive linear term, thus for high threshold $u$ (0 unless input exceeds $t$ and positive after) we have $\text{sgn}(f(\boldsymbol{x})) = \vee \text{sgn}(\boldsymbol{x}^T \boldsymbol{w}_i^* - t)$. This is because, for any $i$, $P$ applied to $X_i > 0$ and $\forall j \neq i, X_j = 0$ gives us $c_i$ which is positive. Also, $P(0) = 0$. Thus, $\text{sgn}(P)$ is 1 if any of the inputs are positive. Using this, we have,

$$Pr[\text{sgn}(f(\boldsymbol{x})) | \boldsymbol{x} \in l(\boldsymbol{z}, t', \epsilon)] \geq Pr[\text{sgn}((\boldsymbol{w}_1^*)^T \boldsymbol{x} - t) | \boldsymbol{x} \in l(\boldsymbol{z}, t', \epsilon)]$$

Also,

$$Pr[\text{sgn}(f(\boldsymbol{x})) | \boldsymbol{x} \in l(\boldsymbol{z}, t', \epsilon)]$$

$$\leq \sum Pr[\text{sgn}(\boldsymbol{x}^T \boldsymbol{w}_i^* - t) | \boldsymbol{x} \in l(\boldsymbol{z}, t', \epsilon)]$$

$$= Pr[\text{sgn}((\boldsymbol{w}_1^*)^T \boldsymbol{x} - t) | \boldsymbol{x} \in l(\boldsymbol{z}, t', \epsilon)] + \sum_{i \neq 1} Pr[\text{sgn}(\boldsymbol{x}^T \boldsymbol{w}_i^* - t) | \boldsymbol{x} \in l(\boldsymbol{z}, t', \epsilon)]$$

$$\leq Pr[\text{sgn}((\boldsymbol{w}_1^*)^T \boldsymbol{x} - t) | \boldsymbol{x} \in l(\boldsymbol{z}, t, \epsilon)] + \eta$$

where $\sum_{i \neq 1} Pr[\text{sgn}(\boldsymbol{x}^T \boldsymbol{w}_i^* - t) | \boldsymbol{x} \in l(\boldsymbol{z}, t', \epsilon)] \leq \eta$. We will show that $\eta$ is not large since a $\boldsymbol{z}$ is close to one of the vectors, it can not be close to the others thus $\alpha_i$ will be large for all $i \neq j$. Let us bound $\eta$,

$$\sum_{i \neq 1} Pr[\text{sgn}(\boldsymbol{x}^T \boldsymbol{w}_i^* - t) | \boldsymbol{x} \in l(\boldsymbol{z}, t', \epsilon)] \leq \sum_{i \neq 1} \left( \Phi^c \left( \frac{t - t_i^* \cos(\alpha_i)}{|\sin(\alpha_i)|} \right) + O(\epsilon) t_i' \right)$$

$$\leq \sum_{i \neq 1} \left( \Phi^c \left( \frac{t - t_i^* \cos(\alpha_i)}{|\sin(\alpha_i)|} \right) + O(\epsilon) t' \right)$$

$$\leq \sum_{i \neq 1} \left( \Phi^c \left( \frac{t - t' \cos(\alpha_i)}{|\sin(\alpha_i)|} \right) + O(\epsilon) t' \right)$$

$$\leq \sum_{i \neq 1} \frac{1}{\sqrt{2\pi}\gamma_i} e^{-\frac{\gamma_i^2}{2}} + O(\epsilon) k t'$$

where $\gamma_i = \frac{t - t' \cos(\alpha_i)}{|\sin(\alpha_i)|}$. The above follows since $\gamma_i \geq 0$ by assumption on $t'$. Under the assumption, let $\beta = \max_{i \neq 1} \cos(\alpha_i)$ we have

$$\gamma_i \geq \frac{t \left( 1 - \frac{\beta}{\cos(\alpha_1)} \right)}{\sqrt{1 - \beta^2}} = \Omega(t)$$

under our setting. Thus we have,

$$\sum_{i \neq 1} Pr[\text{sgn}(\boldsymbol{x}^T \boldsymbol{w}_i^* - t)) | \boldsymbol{x} \in l(\boldsymbol{z}, t', \epsilon)] \leq d e^{-\Omega(t^2)} + O(\epsilon) d t = d e^{-\Omega(t^2)}$$

for small enough $\epsilon$. $\qquad \square$

*Proof of Lemma 6.* Let us assume that $\epsilon < c/t'$ for sufficiently small constant $c$, then we have that

$$0.6 \qquad = Pr[\text{sgn}(f(\boldsymbol{x})) \text{ and } \boldsymbol{x} \in l(\boldsymbol{z}, t_2, \epsilon) | \boldsymbol{x} \in l(\boldsymbol{z}, t_2, \epsilon)]$$
$$\geq Pr[x^T \boldsymbol{w}_1^* \geq t \text{ and } \boldsymbol{x} \in l(\boldsymbol{z}, t_2, \epsilon) | \boldsymbol{x} \in l(\boldsymbol{z}, t_2, \epsilon)]$$
$$\geq \Phi^c \left( \frac{t - t^* \cos(\alpha_1)}{|\sin(\alpha_1)|} \right) - 0.1$$
$$\implies \qquad 0.7 \qquad \geq \Phi^c \left( \frac{t - t^* \cos(\alpha_1)}{|\sin(\alpha_1)|} \right)$$
$$\implies \quad (\Phi^c)^{-1}(0.7) \quad \leq \frac{t - t^* \cos(\alpha_1)}{|\sin(\alpha_1)|}$$
$$\implies \qquad t_2 \qquad \leq \frac{t - (\Phi^c)^{-1}(0.7)\sin(\alpha_1)}{\cos(\alpha_1)} + O(1) \leq \frac{t}{\cos(\alpha)} + O(1)$$

Similarly for $t_1$. Now we need to argue that $t_1, t_2 \geq 0$. Observe that

$$Pr[\text{sgn}(f(\boldsymbol{x})) \text{ and } \boldsymbol{x} \in l(\boldsymbol{z}, 0, \epsilon) | \boldsymbol{x} \in l(\boldsymbol{z}, 0, \epsilon)]$$
$$\leq \sum Pr[x^T \boldsymbol{w}_i^* \geq t \text{ and } \boldsymbol{x} \in l(\boldsymbol{z}, 0, \epsilon) | \boldsymbol{x} \in l(\boldsymbol{z}, 0, \epsilon)]$$
$$= \sum \Phi^c \left( \frac{t - \epsilon \cos(\alpha_1)}{|\sin(\alpha_1)|} \right) + O(\epsilon^2) d \leq d e^{-\Omega(t^2)} < 0.4$$

Thus for sufficiently large $t = \Omega(\sqrt{\log d})$, this will be less than 0.4. Hence there will be some $t_1, t_2 \geq 0$ with probability evaluating to 0.4 since the probability is an almost increasing function of $t$ up to small noise in the given range (see proof of Lemma 5). $\qquad \square$

*Proof of Lemma 7.* Let $V$ be the plane spanned by $\boldsymbol{w}_1^*$ and $\boldsymbol{z}$ and let $\boldsymbol{v}_1 = \boldsymbol{w}_1^*$ and $\boldsymbol{v}_2$ be the basis of this space. Thus, we can write $\boldsymbol{z} = \cos(\alpha)\boldsymbol{v}_1 + \sin(\alpha)\boldsymbol{v}_2$.

Let us apply a Gaussian perturbation $\rho$ along the tangential hyperplane normal to $\boldsymbol{z}$. Say it has distribution $\epsilon N(0, 1)$ along any direction tangential to the vector $\boldsymbol{z}$. Let $\epsilon_1$ be the component of $\rho$ on to $V$ and let $\epsilon_2$ be the component perpendicular to it. We can write the perturbation as $\rho = \epsilon_1(\sin(\alpha)\boldsymbol{v}_1 - \cos(\alpha)\boldsymbol{v}_2) + \epsilon_2 \boldsymbol{v}_3$ where $\boldsymbol{v}_3$ is orthogonal to both $\boldsymbol{v}_1$ and $\boldsymbol{v}_2$.

So the new angle $\alpha'$ of $\boldsymbol{z}$ after the perturbation is given by

$$\cos(\alpha') = \frac{\boldsymbol{v}_1^T(\boldsymbol{z} + \rho)}{||\boldsymbol{z} + \rho||}$$
$$= \frac{\cos(\alpha) + \epsilon_1 \sin(\alpha)}{\sqrt{1 + ||\rho||^2}}$$

Note that with constant probability $\epsilon_1 \geq \epsilon$ as $\rho$ is a Gaussian variable with standard deviation $\epsilon$. And with high probability $||\rho|| < O(\epsilon\sqrt{d-1})$. We will set $\epsilon = \Theta(\sin(\alpha)/d) = \Theta(\alpha/d)$. Thus with constant probability:

$$\cos(\alpha') \geq \frac{\cos(\alpha) + \epsilon \sin(\alpha)}{\sqrt{1 + O(\epsilon^2 d)}}$$
$$\geq (\cos(\alpha) + \epsilon \sin(\alpha))(1 - O(\epsilon^2 d))$$
$$\geq \cos(\alpha) + \Omega(\epsilon \sin(\alpha)) - O(\epsilon^2 d)$$
$$\geq \cos(\alpha) + \Omega(\epsilon \sin(\alpha)).$$

Thus change in $\cos(\alpha)$ is given by $\Delta \cos(\alpha) \geq \Omega(\epsilon \sin(\alpha))$. Now change in the angle $\alpha$ satisfies by the Mean Value Theorem:

$$\Delta \cos(\alpha) = \Delta \alpha \left[ \frac{d}{dx} \cos(x) \right]_{x \in [\alpha, \alpha']}$$
$$\implies -\Delta \alpha = \Delta \cos(\alpha) \left[ \frac{1}{\sin(x)} \right]_{x \in [\alpha, \alpha']}$$

$$\geq \frac{\Omega(\epsilon \sin(\alpha))}{\sin(\alpha)} = \Omega(\epsilon) = \Omega(\alpha/d).$$

$\square$

### B.6 LEARNING UNION OF HALFSPACES FAR FROM THE ORIGIN

**Theorem 12.** *Given non-noisy labels from a union of halfspaces that are at a distance $\Omega(\sqrt{\log d})$ and are each a constant angle apart, there is an algorithm to recover the underlying weights to $\epsilon$ closeness in polynomial time.*

*Proof.* Observe that $\bigvee X_i$ is equivalent to $P(X_1, \cdot, X_d) = 1 - \prod(1 - X_i)$. Thus

$$f(\boldsymbol{x}) = \bigvee \mathrm{sgn}(\boldsymbol{x}^T \boldsymbol{w}_i^* - t) = 1 - \prod(1 - \mathrm{sgn}(\boldsymbol{x}^T \boldsymbol{w}_i^* - t)).$$

Since $P$ and sgn here satisfies our assumptions 1, 2, for $t = \Omega(\sqrt{\log d})$ (see Lemma 11) we can apply Theorem 11 to recover the vectors $\boldsymbol{w}_i^*$ approximately. Subsequently, refining to arbitrarily close using Theorem 5 is possible due to the monotonicity. Thus we can recover the vectors to arbitrary closeness in polynomial time. $\square$

### B.7 SIGMOID ACTIVATIONS

Observe that for sigmoid activation, Assumption 2 is satisfied for $\rho(t, \sigma) = e^{-t+\sigma^2/2}$. Thus to satisfy Assumption 3, we need $t = \Omega(\eta \log d)$.

Note that for such value of $t$, the probability of the threshold being crossed is small. To avoid this we further assume that $f$ is non-negative and we have access to an oracle that biases the samples towards larger values of $f$; that after $\boldsymbol{x}$ is drawn from the Gaussian distribution, it retains the sample $(\boldsymbol{x}, f(\boldsymbol{x}))$ with probability proportional to $f(\boldsymbol{x})$ – so $Pr[\boldsymbol{x}]$ in the new distribution. This enables us to compute correlations even if $E_{\boldsymbol{x}\tilde{N}(0,\mathbf{I})}[f(\boldsymbol{x})]$ is small. In particular by computing $\mathbb{E}[h(\boldsymbol{x})]$ from this distribution, we are obtaining $\mathbb{E}[f(\boldsymbol{x})h(\boldsymbol{x})]/\mathbb{E}[f(\boldsymbol{x})]$ in the original distribution. Thus we can compute correlations that are scaled.

We get our approximate theorem:

**Theorem 13.** *For $t = \Omega(\log d)$, columns of $(\mathbf{TW}^*)^{-1}$ can be recovered within error $1/\mathsf{poly}(d)$ using the algorithm in polynomial time.*

### B.8 POLYNOMIALS $P$ WITH HIGHER DEGREE IN ONE VARIABLE

In the main section we assumed that the polynomial has degree at most 1 in each variable. Let us give a high level overview of how to extend this to the case where each variable is allowed a large degree. $P$ now has the following structure,

$$P(X_1, \ldots, X_d) = \sum_{\boldsymbol{r} \in \mathbb{Z}_+^d} c_{\boldsymbol{r}} \prod_{i=1}^d X_i^{r_i}$$

If $P$ has a higher degree in $X_i$ then Assumption 2 changes to a more complex (stronger) condition. Let $q_i(x) = \sum_{\boldsymbol{r} \in \mathbb{Z}_+^d | \forall j \neq i, r_j = 0} c_{\boldsymbol{r}} x^{r_i}$, that is $q_i$ is obtained by setting all $X_j$ for $j \neq i$ to 0.

**Assumption 4.** $\mathbb{E}_{g \sim N(0,\sigma^2)}[|u_t(g)|^r] \leq \rho(t, \sigma)$ *for all* $r \in \mathbb{Z}_+$[8]. $\mathbb{E}[q_i(u_t(g))h_k(g))] = t^{\Theta(1)}\rho(t, 1)$ *for* $k = 2, 4$. *Lastly, for all* $d \geq i > 1$, $\sum_{\substack{\boldsymbol{r} \in \mathbb{Z}_+^d \\ ||\boldsymbol{r}||_0 = i}} |c_{\boldsymbol{r}}| \leq d^{O(i)}$.

The last assumption holds for the case when the degree is a constant and each coefficient is upper bounded by a constant. It can hold for decaying coefficients.

Let us collect the univariate terms $P_{\mathsf{uni}}(X) = \sum_{i=1}^d q_i(X_i)$. Corresponding to the same we get $f_{\mathsf{uni}}$. This will correspond to the $f_{\mathsf{lin}}$ we had before. Note that the difference now is that instead of

---

[8]For example, his would hold for any $u$ bounded in $[-1, 1]$ such as sigmoid or sign.

being the same activation for each weight vector, now we have different ones $q_i$ for each. Using $H_4$ correlation as before, now we get that:

$$\mathbb{E}[f_{\mathsf{uni}}(\boldsymbol{x})H_4(\boldsymbol{z}^T\boldsymbol{x})] = \sum_{i=1}^{d} \widehat{q_i \circ u_t}_4 (\boldsymbol{z}^T\boldsymbol{w}_i^*)^4 \quad \text{and} \quad \mathbb{E}[f_{\mathsf{uni}}(\boldsymbol{x})H_2(\boldsymbol{z}^T\boldsymbol{x})] = \sum_{i=1}^{d} \widehat{q_i \circ u_t}_2 (\boldsymbol{z}^T\boldsymbol{w}_i^*)^2$$

where $\widehat{q_i \circ u_t}$ are hermite coefficients for $q_i \circ u_t$. Now the assumption guarantees that these are positive which is what we had in the degree 1 case.

Second we need to show that even with higher degree, $\mathbb{E}[|f(\boldsymbol{x}) - f_{\mathsf{uni}}(\boldsymbol{x})|]$ is small. Observe that

**Lemma 17.** *For $\boldsymbol{r}$ such that $||\boldsymbol{r}||_0 > 1$, under Assumption 4 we have,*

$$\mathbb{E}\left[\prod_{i=1}^{d} \left(u_t((\boldsymbol{w}_j^*)^T\boldsymbol{x})\right)^{r_i}\right] \le \rho(t,1)O\left(\rho(t,||\mathbf{W}^*||)\right)^{||\boldsymbol{r}||_0-1}.$$

The proof essentially uses the same idea, except that now the dependence is not on $||\boldsymbol{r}||_1$ but only the number of non-zero entries (number of different weight vectors). With this bound, we can now bound the deviation in expectation.

**Lemma 18.** *Let $\Delta(\boldsymbol{x}) = f(\boldsymbol{x}) - f_{\mathsf{uni}}(\boldsymbol{x})$. Under Assumptions 4, if $t$ is such that $\rho(t,||\mathbf{W}^*||) \le d^{-C}$ for large enough constant $C > 0$, we have, $\mathbb{E}[|\Delta(\boldsymbol{x})|] \le d^{O(1)}\rho(t,1)\rho(t,||\mathbf{W}^*||)$.*

*Proof.* We have,

$$\mathbb{E}[|\Delta(\boldsymbol{x})|] = \mathbb{E}\left[\left|\sum_{\substack{\boldsymbol{r}\in\mathbb{Z}_+^d \\ r_i \le D, ||\boldsymbol{r}||_0 > 1}} c_{\boldsymbol{r}} \prod_{i=1}^{d} \left(u_t((\boldsymbol{w}_j^*)^T\boldsymbol{x})\right)^{r_i}\right|\right]$$

$$\le \sum_{\substack{\boldsymbol{r}\in\mathbb{Z}_+^d \\ r_i \le D, ||\boldsymbol{r}||_0 > 1}} |c_{\boldsymbol{r}}|\mathbb{E}\left[\left|\prod_{i=1}^{d} \left(u_t((\boldsymbol{w}_j^*)^T\boldsymbol{x})\right)^{r_i}\right|\right]$$

$$= \sum_{\substack{\boldsymbol{r}\in\mathbb{Z}_+^d \\ r_i \le D, ||\boldsymbol{r}||_0 > 1}} |c_{\boldsymbol{r}}|\rho(t,1)\left(\rho(t,||\mathbf{W}^*||)\right)^{||\boldsymbol{r}||_0-1}$$

$$\le C\sum_{i=1}^{d} \binom{d}{i} D^i \rho(t,1)\left(\rho(t,||\mathbf{W}^*||)\right)^{i-1}$$

$$\le d^C \sum_{i=1}^{d} \rho(t,1)\left(d^C\rho(t,||\mathbf{W}^*||)\right)^{i-1} \qquad\qquad \text{using Assumption 4}$$

$$\le d^{2C+1}\rho(t,1)\rho(t,||\mathbf{W}^*||) \qquad\qquad \text{since } \rho(t,||\mathbf{W}^*||) \le d^{-C}.$$

$\square$

Thus as before, if we choose $t$ appropriately, we get the required results. Similar ideas can be used to extend to non-constant degree under stronger conditions on the coefficients.

## C   SAMPLE COMPLEXITY

*Proof of Lemma 3.* $C_1(f, \boldsymbol{z}, s) = E[f(\boldsymbol{x})\delta(\boldsymbol{z}^T\boldsymbol{x} - s)] = \int_{\boldsymbol{x}} f(\boldsymbol{x})\delta(\boldsymbol{z}^T\boldsymbol{x} - s)\phi(\boldsymbol{x})d\boldsymbol{x}$ Let $x_0$ be the component of $\boldsymbol{x}$ along $\boldsymbol{z}$ and $y$ be the component along $\boldsymbol{z}^\perp$. So $\boldsymbol{x} = x_0\hat{\boldsymbol{z}} + y\boldsymbol{z}^\perp$. Interpreting $\boldsymbol{x}$ as a function of $x_0$ and $y$:

$$C_1(f, \boldsymbol{z}, s) = \int_y \int_{x_0} f(\boldsymbol{x})\delta(x_0 - s)\phi(x_0)\phi(y)dx_0 dy$$

$$= \int_y [f(\boldsymbol{x})]_{x_0=s}\phi(y)dy$$

$$= \phi(s)E[f(\boldsymbol{x})|x_0 = s]$$

$$= \phi(\boldsymbol{z}^T\boldsymbol{x} = s)E[f(\boldsymbol{x})|x_0 = s]$$

where the second equality follows from $\int_x \delta(x-a)f(x) = [f(x)]_{x=a}$. $\square$

*Proof of Claim 4.* Let $x_0$ be the component of $\boldsymbol{x}$ along $\boldsymbol{z}$ and $y$ be the component of $\boldsymbol{x}$ in the space orthogonal to $\boldsymbol{z}$. Let $\hat{\boldsymbol{z}}$ denote a unit vector along $\boldsymbol{z}$. We have $\boldsymbol{x} = x_0\hat{\boldsymbol{z}} + y$ and $\frac{\partial \boldsymbol{x}}{\partial x_0} = \hat{\boldsymbol{z}}$. So, correlation can be computed as follows:

$$\mathbb{E}[P[\boldsymbol{x}].\delta'(\boldsymbol{z}^T\boldsymbol{x}-s)] = \int_{\boldsymbol{y}} \phi(\boldsymbol{y})\int_{x_0}\delta'(x_0-s)P[\boldsymbol{x}]\phi(x_0)dx_0d\boldsymbol{y}$$

Since $\int_x \delta'(x-a)f(x)dx = [\frac{df}{dx}](x=a)$ this implies:

$$\mathbb{E}[P[\boldsymbol{x}]\delta'(\boldsymbol{z}^T\boldsymbol{x}-s)]$$

$$= \int_{\boldsymbol{y}} \left[\frac{\partial}{\partial x_0}P[\boldsymbol{x}]\phi(\boldsymbol{x})\right]_{x_0=s} d\boldsymbol{y}$$

$$= \int_{\boldsymbol{y}} \left[\phi(x_0)\sum_i \frac{\partial P}{\partial X_i}\cdot\frac{\partial X_i}{\partial x_0} + P[\boldsymbol{x}]\phi'(x_0))\right]_{x_0=s} \phi(\boldsymbol{y})d\boldsymbol{y}$$

$$= \sum_i \int_{\boldsymbol{y}} \left[\phi(x_0)\frac{\partial P}{\partial X_i}\cdot\frac{\partial X_i}{\partial x_0}\right]_{x_0=s} \phi(\boldsymbol{y})d\boldsymbol{y} + \phi'(s)\int_{\boldsymbol{y}} P[\boldsymbol{x}]\phi(\boldsymbol{y})d\boldsymbol{y}$$

Note that $\frac{\partial X_i}{\partial x_0} = \frac{\partial}{\partial x_0}u(\boldsymbol{x}^T\boldsymbol{w}_i^* - t) = u'(\boldsymbol{x}^T\boldsymbol{w}_i^* - t)\hat{\boldsymbol{z}}^T\boldsymbol{w}_i^*$. If $u$ is the sign function then $u'(x) = \delta(x)$. So focusing on one summand in the sum we get

$$\int_{\boldsymbol{y}} \left[\phi(x_0)\frac{\partial P}{\partial X_i}\cdot\frac{\partial X_i}{\partial x_0}\right]_{x_0=s} \phi(\boldsymbol{y})d\boldsymbol{y}$$

$$= \int_{\boldsymbol{y}} [\phi(x_0)u'(\boldsymbol{x}^T\boldsymbol{w}_i^* - t)(\hat{\boldsymbol{z}}^T\boldsymbol{w}_i^*)\frac{\partial P}{\partial X_i}]_{x_0=s}\phi(\boldsymbol{y})d\boldsymbol{y}$$

$$= \int_{\boldsymbol{y}} (\hat{\boldsymbol{z}}^T\boldsymbol{w}_i^*)[\phi(x_0)\delta(\hat{\boldsymbol{z}}^T\boldsymbol{w}_i^*x_0 + (\boldsymbol{w}_i^*)^Ty - t)\frac{\partial P}{\partial X_i}]_{x_0=s}\phi(\boldsymbol{y})d\boldsymbol{y}$$

$$= (\hat{\boldsymbol{z}}^T\boldsymbol{w}_i^*)\int_{\boldsymbol{y}} \phi(s)\delta(s(\boldsymbol{w}_1^*)^T\hat{\boldsymbol{z}} + (\boldsymbol{w}_1^*)^Ty - t)\frac{\partial P}{\partial X_i}\phi(\boldsymbol{y})d\boldsymbol{y}$$

Again let $y = y_0(\boldsymbol{w}_i^*)' + z$ where $z$ is perpendicular to $\boldsymbol{w}_i^*$ and $\boldsymbol{z}$. And $(\boldsymbol{w}_i^*)'$ is perpendicular component of $\boldsymbol{w}_i^*$ to $\boldsymbol{z}$. Interpreting $\boldsymbol{x} = t\hat{\boldsymbol{z}} + y_0(\boldsymbol{w}_1^*)' + z$ as a function of $y_0, z$ we get:

$$= (\hat{\boldsymbol{z}}^T\boldsymbol{w}_i^*)\int_z\int_{y_0} \phi(s)\delta(s(\boldsymbol{w}_1^*)^T\hat{\boldsymbol{z}} + ((\boldsymbol{w}_1^*)^T(\boldsymbol{w}_1^*)')y_0 - t)\phi(y_0)\phi(z)\frac{\partial P}{\partial X_i}dy_0dz$$

Note that by substituting $v = ax$ we get $\int_{x=-\infty}^{\infty} f(x)\delta(ax-b)dx = \int_{x=-\infty/a}^{\infty/a} f(x)\delta(ax-b)dx = \text{sgn}(a)\frac{1}{a}f(\frac{b}{a}) = \frac{1}{|a|}f(\frac{b}{a})$. So this becomes:

$$= \frac{\hat{\boldsymbol{z}}^T\boldsymbol{w}_i^*}{|(\boldsymbol{w}_i^*)^T(\boldsymbol{w}_i^*)'|}\int_z \phi(s)[\phi(y_0)\frac{\partial P}{\partial X_i}]_{y_0=\frac{t-s\hat{\boldsymbol{z}}^T\boldsymbol{w}_i^*}{(\boldsymbol{w}_i^*)^T(\boldsymbol{w}_i^*)'}}\phi(z)dz$$

$$= \frac{\hat{\boldsymbol{z}}^T \boldsymbol{w}_i^*}{|(\boldsymbol{w}_i^*)^T (\boldsymbol{w}_i^*)'|} \int_z [\phi(y_0)\phi(x_0)\phi(z)\frac{\partial P}{\partial X_i}]_{x_0=s, y_0 = \frac{t - s\hat{\boldsymbol{z}}^T \boldsymbol{w}_i^*}{(\boldsymbol{w}_i^*)^T (\boldsymbol{w}_i^*)'}} \, dz$$

$$= \frac{\hat{\boldsymbol{z}}^T \boldsymbol{w}_i^*}{|(\boldsymbol{w}_i^*)^T (\boldsymbol{w}_i^*)'|} \phi\left(y_0 = \frac{t - s\hat{\boldsymbol{z}}^T \boldsymbol{w}_i^*}{(\boldsymbol{w}_i^*)^T (\boldsymbol{w}_i^*)'}\right)\phi(x_0 = t)\int_z [\phi(z)\frac{\partial P}{\partial X_i}]_{x_0=t, y_0 = \frac{t - s\hat{\boldsymbol{z}}^T \boldsymbol{w}_i^*}{(\boldsymbol{w}_i^*)^T (\boldsymbol{w}_i^*)'}} \, dz$$

$$= \frac{\hat{\boldsymbol{z}}^T \boldsymbol{w}_i^*}{|(\boldsymbol{w}_i^*)^T (\boldsymbol{w}_i^*)'|} \int_z [\phi(\boldsymbol{x})\frac{\partial P}{\partial X_i}]_{x_0=t, y_0 = \frac{t - s\hat{\boldsymbol{z}}^T \boldsymbol{w}_i^*}{(\boldsymbol{w}_i^*)^T (\boldsymbol{w}_i^*)'}} \, dz$$

$$= \frac{\hat{\boldsymbol{z}}^T \boldsymbol{w}_i^*}{|(\boldsymbol{w}_i^*)^T (\boldsymbol{w}_i^*)'|} \int_{\boldsymbol{x}: \boldsymbol{z}^T \boldsymbol{x}=s, \boldsymbol{x}^T \boldsymbol{w}_i^*=t} \phi(\boldsymbol{x})\frac{\partial P}{\partial X_i} \, d\boldsymbol{x}$$

$$= \frac{\hat{\boldsymbol{z}}^T \boldsymbol{w}_i^*}{|(\boldsymbol{w}_i^*)^T (\boldsymbol{w}_i^*)'|} \phi(\boldsymbol{z}^T \boldsymbol{x}=s, \boldsymbol{x}^T \boldsymbol{w}_i^*=t)\mathbb{E}[\frac{\partial P}{\partial X_i}|\boldsymbol{x}^T \boldsymbol{z}=s, \boldsymbol{x}^T \boldsymbol{w}_i^*=t].$$

Let $\alpha_i$ be the angle between $\boldsymbol{z}$ and $\boldsymbol{w}_i^*$. Then this is

$$= |\cot(\alpha_i)|\phi(\boldsymbol{z}^T \boldsymbol{x}=t, (\boldsymbol{w}_i^*)^T \boldsymbol{x}=t)\mathbb{E}[\frac{\partial P}{\partial X_i}]|\boldsymbol{x}^T \boldsymbol{z}=s, \boldsymbol{x}^T \boldsymbol{w}_i^*=t]$$

Thus, overall correlation

$$= \sum_{i \in S} |\cot(\alpha_i)|\phi(\boldsymbol{z}^T \boldsymbol{x}=s, \boldsymbol{x}^T \boldsymbol{w}_i^*=t)\mathbb{E}[\frac{\partial P}{\partial X_i}|\boldsymbol{x}^T \boldsymbol{z}=s, \boldsymbol{x}^T \boldsymbol{w}_i^*=t]$$
$$+ \phi'(s)\mathbb{E}[P[\boldsymbol{x}]|\boldsymbol{x}^T \boldsymbol{z}=s]$$

$\square$

*Proof of Claim 5.* Note that for small $\alpha$, $|\cot \alpha| = O(1/\alpha) \leq O(1/\epsilon_2)$. Since $P(X) \leq ||X||^{c_1}$, we have $f(\boldsymbol{x}) \leq d^{c_1}$ (as with sgn function each $\text{sgn}((\boldsymbol{w}_i^*)^T \boldsymbol{x}) \leq 1$) and all $Q_i[\boldsymbol{x}], R_i[\boldsymbol{x}]$ are at most $2d^{c_1}$.

By Cauchy's mean value theorem $C_2(f, \boldsymbol{z}, t) = 2\epsilon[\mathbb{E}[f(\boldsymbol{x})\delta'(\boldsymbol{z}^T \boldsymbol{x} = s)]]_{s \in t \pm \epsilon}$. Note that $\cot(\alpha_i)\phi(\boldsymbol{z}^T \boldsymbol{x} = t, (\boldsymbol{w}_i^*)^T \boldsymbol{x} = t) = \cot(\alpha_i)\phi(t\tan(\alpha_i/2))$ which is a decreasing function of $\alpha_i$ in the range $[0, \pi]$ So if all $\alpha_i$ are upper bounded by $\epsilon_2$ then by above corollary,

$$C_2(f, \boldsymbol{z}, s) \leq 2\epsilon n \cot(\epsilon_2)\phi(\boldsymbol{z}^T \boldsymbol{x} = t, (\boldsymbol{w}_1^*)^T \boldsymbol{x} = t)(2d^{c_1}) + (2d^{c_1})$$
$$= 2\epsilon n \cot(\epsilon_2)\phi(t\tan(\epsilon_2/2))(2d^{c_1}) + (2d^{c_1})$$
$$\leq \frac{\epsilon}{\epsilon_2} d^{O(1)}.$$

$\square$

Observe that the above proof does not really depend on $P$ and holds for for any polynomial of $u((\boldsymbol{w}_i^*)^T \boldsymbol{x})$ as long as the polynomial is bounded and the $\boldsymbol{w}_i^*$ are far off from $\boldsymbol{z}$.

*Proof Of Lemma 8.* If $\boldsymbol{z} = \boldsymbol{w}_i^*$, then

$$C_2(f, \boldsymbol{z}, t) = E[f(\boldsymbol{x})(\delta(\boldsymbol{z}^T \boldsymbol{x} - t - \epsilon) - \delta(\boldsymbol{z}^T \boldsymbol{x} - t + \epsilon))]$$
$$= E[u((\boldsymbol{w}_i^*)^T \boldsymbol{x})Q_i[\boldsymbol{x}](\delta(\boldsymbol{z}^T \boldsymbol{x} - t - \epsilon) - \delta(\boldsymbol{z}^T \boldsymbol{x} - t + \epsilon))]$$
$$+ E[R_i[\boldsymbol{x}](\delta(\boldsymbol{z}^T \boldsymbol{x} - t - \epsilon) - \delta(\boldsymbol{z}^T \boldsymbol{x} - t + \epsilon))]$$

Since $u((\boldsymbol{w}_i^*)^T \boldsymbol{x}) = 0$ for $\boldsymbol{z}^T \boldsymbol{x} = t - \epsilon$ and 1 for $\boldsymbol{z}^T \boldsymbol{x} = t + \epsilon$, and using the Cauchy mean value theorem for the second term this is

$$= E[Q_i[\boldsymbol{x}]\delta(\boldsymbol{z}^T \boldsymbol{x} - t - \epsilon)] + 2\epsilon[\mathbb{E}[R_i[\boldsymbol{x}]\delta'(\boldsymbol{z}^T \boldsymbol{x} - s_1)]]_{s_1 \in t \pm \epsilon}$$
$$= E[Q_i[\boldsymbol{x}]\delta(\boldsymbol{z}^T \boldsymbol{x} - t)] + E[R_i[\boldsymbol{x}](\delta(\boldsymbol{z}^T \boldsymbol{x} - t - \epsilon) - \delta(\boldsymbol{z}^T \boldsymbol{x} - t))]$$
$$= \phi(t)E[Q_i[\boldsymbol{x}]|\boldsymbol{z}^T \boldsymbol{x} = t + \epsilon] + \epsilon[C_2(Q_i, \boldsymbol{z}, s_2)]_{s_2 \in [t, t+\epsilon]} + 2\epsilon[C_2(R_i, \boldsymbol{z}, s)]_{s \in t \pm \epsilon}$$

$$= \phi(t)E[Q_i[\boldsymbol{x}]|\boldsymbol{z}^T\boldsymbol{x} = t] + \epsilon d^{O(1)}$$

The last step follows from Claim 5 applied on $Q_i$ and $R_i$ as all the directions of $\boldsymbol{w}_j^*$ are well separated from $\boldsymbol{z} = \boldsymbol{w}_i^*$ and $\boldsymbol{w}_i^*$ is absent from both $Q_i$ and $R_i$. Also the corresponding $Q_i$ and $R_i$ are bounded. □

## C.1 ReLU activation

If $u$ is the RELU activation, the high level idea is to use correlation with the second derivative $\delta''$ of the Dirac delta function instead of $\delta'$. More precisely we will compute $C_3(f, \boldsymbol{z}, s) = \mathbb{E}[f.(\delta'(\boldsymbol{z}^T\boldsymbol{x} - s - \epsilon) - \delta'(\boldsymbol{z}^T\boldsymbol{x} - s + \epsilon)]$. Although we show the analysis only for the RELU activation, the same idea works for any activation that has non-zero derivative at 0.

Note that now $u' = \text{sgn}$ and $u'' = \delta$.

For ReLU activation, Lemma 8 gets replaced by the following Lemma. The rest of the argument is as for the sgn activation. We will need to assume that $P$ has constant degree and sum of absolute value of all coefficients is $\text{poly}(d)$

**Lemma 19.** *Assuming polynomial $P$ has constant degree, and sum of the magnitude of all coefficients is at most $\text{poly}(d)$, if $\boldsymbol{z} = \boldsymbol{w}_i^*$ then $C_3(f, \boldsymbol{z}, t) = \mathbb{E}[\phi(t)\frac{\partial}{\partial X_i}P + \phi(t)\sum_{j \neq i}\cos(\alpha_j)sgn(\boldsymbol{x}^T\boldsymbol{w}_i^* - t)\frac{\partial}{\partial X_j}P + \phi'(t)P|\boldsymbol{x}^T\boldsymbol{w}_i^* = t] + \epsilon d^{O(1)}$. Otherwise if all angles $\alpha_i$ between $\boldsymbol{z}$ and $\boldsymbol{w}_i$ are at least $\epsilon_2$ it is at most $\epsilon d^{O(1)}/\epsilon_2$.*

We will prove the above lemma in the rest of this section. First we will show that $\boldsymbol{z}$ is far from any of the $\boldsymbol{w}_i^*$'s then $\mathbb{E}[P.\delta''(\boldsymbol{z}^T\boldsymbol{x} - s)]$ is bounded.

**Lemma 20.** *If the sum of the absolute value of the coefficients of $P$ is bounded by $\text{poly}(d)$, its degree is at most constant, $\alpha_i > \epsilon_2$ then $\mathbb{E}[P.\delta''(\boldsymbol{z}^T\boldsymbol{x} - s)]$ is $d^{O(1)}/\epsilon_2$.*

*Proof.* Let $x_0$ be the component of $\boldsymbol{x}$ along $\boldsymbol{z}$ and $y$ be the component of $\boldsymbol{x}$ in the space orthogonal to $\boldsymbol{z}$ as before. We have $\boldsymbol{x} = x_0\hat{\boldsymbol{z}} + \boldsymbol{y}$ and $\frac{\partial \boldsymbol{x}}{\partial x_0} = \hat{\boldsymbol{z}}$. We will look at monomials $M_l$ in $P = \sum_l M_l$. As before since $\int_x \delta''(x - a)f(x)dx = \left[\frac{d^2f}{dx^2}\right]_{x=a}$ we get

$$\mathbb{E}[f(\boldsymbol{x}).\delta''(\boldsymbol{z}^T\boldsymbol{x} - s)] = \int_{\boldsymbol{y}} f(\boldsymbol{x})\phi(\boldsymbol{x})\delta''(\boldsymbol{z}^T\boldsymbol{x} - s)d\boldsymbol{y}$$

$$= \int_{\boldsymbol{y}} \left[\frac{\partial^2}{\partial x_0^2}(P[\boldsymbol{x}]\phi(\boldsymbol{x}))\right]_{x_0=s} d\boldsymbol{y}$$

$$= \sum_l \int_{\boldsymbol{y}} \left[\frac{\partial^2}{\partial x_0^2}(M_l[\boldsymbol{x}]\phi(x_0))\right]_{x_0=s} \phi(\boldsymbol{y})d\boldsymbol{y}$$

Now consider a monomial $M = X_1^{i_1}..X_k^{i_k}$.

Take the symbolic second derivative $\frac{\partial^2}{\partial x_0^2}$ of $M[\boldsymbol{x}]\phi(x_0)$ w.r.t $x_0$. This will produce a polynomial involving $X_i$'s, $\frac{\partial X_i}{\partial x_0}$, $\frac{\partial^2 X_i}{\partial x_0^2}$, $\phi, \phi', \phi''$. Let us examine each of these terms.

$$\frac{\partial}{\partial x_0}X_i[\boldsymbol{x}] = \frac{\partial}{\partial x_0}u(\boldsymbol{x}^T\boldsymbol{w}_i^* - t)$$

$$= \text{sgn}(\boldsymbol{x}^T\boldsymbol{w}_i^* - t)(\boldsymbol{w}_i^*)^T\frac{\partial}{\partial x_0}\boldsymbol{x}$$

$$= \text{sgn}(\boldsymbol{x}^T\boldsymbol{w}_i^* - t)((\boldsymbol{w}_i^*)^T\boldsymbol{z})$$

$$= \cos(\alpha_i)\text{sgn}(\boldsymbol{x}^T\boldsymbol{w}_i^* - t)$$

Thus $\frac{\partial}{\partial x_0}X_i[\boldsymbol{x}]$ is a bounded function of $\boldsymbol{x}$. We have

$$\frac{\partial^2}{\partial x_0^2}X_i(\boldsymbol{x}) = \frac{\partial}{\partial x_0}\text{sgn}(\boldsymbol{x}^T\boldsymbol{w}_i^* - t)((\boldsymbol{w}_i^*)^T\boldsymbol{z}) = \cos^2(\alpha_i)\delta(\boldsymbol{x}^T\boldsymbol{w}_i^* - t).$$

Again as before

$$\int_{\boldsymbol{y}} [\delta(\boldsymbol{x}^T\boldsymbol{w}_i^* - t)g(\boldsymbol{x})\phi(x_0)\phi(\boldsymbol{y})]_{x_0=s}d\boldsymbol{y}$$
$$= (1/|\sin(\alpha_i)|)\mathbb{E}[g(\boldsymbol{x})|x_0 = s, \boldsymbol{x}^T\boldsymbol{w}_i^* = t]\phi(x_0 = s, \boldsymbol{x}^T\boldsymbol{w}_i^* = t)$$

Note that if the degree is bounded, since $|\sin(\alpha_i)|$ is at least $\epsilon_2$ expected value of each monomial obtained is bounded. So the total correlation is $\text{poly}(d)/\epsilon_2$. □

*Proof of Lemma 20.* As in the case of sgn activation, if $\boldsymbol{z} = \boldsymbol{w}_i^*$,

$$\mathbb{E}[P[\boldsymbol{x}]\delta'(\boldsymbol{x}^T\boldsymbol{w}_i^* - s)]$$
$$= \int_{\boldsymbol{x}} P[\boldsymbol{x}]\phi(\boldsymbol{x})\delta'(\boldsymbol{z}^T\boldsymbol{x} - s)d\boldsymbol{x}$$
$$= \int_{\boldsymbol{y}} \left[\frac{\partial}{\partial x_0}(P[\boldsymbol{x}]\phi(\boldsymbol{x}))\right]_{x_0=s} d\boldsymbol{y}$$
$$= \int_{\boldsymbol{y}} \left[\sum_j \left(\frac{\partial P}{\partial X_j}\right)[\boldsymbol{x}]\frac{\partial X_j[\boldsymbol{x}]}{\partial x_0}\phi(\boldsymbol{x}) + P[\boldsymbol{x}]\frac{\partial\phi(\boldsymbol{x})}{\partial x_0}\right]_{x_0=s} d\boldsymbol{y}$$
$$= \int_{\boldsymbol{y}} \phi(\boldsymbol{y}) \left[\text{sgn}(x_0 - t)\phi(s)\left(\frac{\partial P}{\partial X_i}\right)[\boldsymbol{x}] + \sum_{j\neq i}\text{sgn}(\boldsymbol{x}^T\boldsymbol{w}_j^* - t)\cos(\alpha_j)\phi(s)\left(\frac{\partial P}{\partial X_j}\right)[\boldsymbol{x}] + P\phi'(x_0)\right]_{x_0=s} d\boldsymbol{y}$$
$$= \int_{\boldsymbol{y}} \phi(\boldsymbol{y}) \left[\text{sgn}(s - t)\phi(s)\left(\frac{\partial P}{\partial X_i}\right)[\boldsymbol{x}] + \sum_{j\neq i}\text{sgn}(\boldsymbol{x}^T\boldsymbol{w}_j^* - t)\cos(\alpha_j)\phi(s)\left(\frac{\partial P}{\partial X_j}\right)[\boldsymbol{x}] + P\phi'(x_0)\right]_{x_0=s} d\boldsymbol{y}$$

For $s = t + \epsilon$ this is

$$= \int_{\boldsymbol{y}} \phi(\boldsymbol{y}) \left[\phi(s)\left(\frac{\partial P}{\partial X_i}\right)[\boldsymbol{x}] + \sum_{j\neq i}\cos(\alpha_j)\text{sgn}(\boldsymbol{x}^T\boldsymbol{w}_j^* - t)\phi(s)\left(\frac{\partial P}{\partial X_j}\right)[\boldsymbol{x}] + P\phi'(s)\right]_{s=t+\epsilon} d\boldsymbol{y}$$
$$= \left(\int_{\boldsymbol{y}} \phi(\boldsymbol{y}) \left(\phi(t)\left(\frac{\partial P}{\partial X_i}\right)[\boldsymbol{x}] + \sum_{j\neq i}\cos(\alpha_j)\text{sgn}(\boldsymbol{x}^T\boldsymbol{w}_j^* - t)\phi(t)\left(\frac{\partial P}{\partial X_j}\right)[\boldsymbol{x}] + P\phi'(t)\right) d\boldsymbol{y}\right) + \epsilon d^{O(1)}$$
$$= \mathbb{E}\left[\phi(t)\left(\frac{\partial P}{\partial X_i}\right)[\boldsymbol{x}] + \phi(t)\sum_{j\neq i}\cos(\alpha_j)\text{sgn}(\boldsymbol{x}^T\boldsymbol{w}_i^* - t)\left(\frac{\partial P}{\partial X_j}\right)[\boldsymbol{x}] + \phi'(t)P \Big| \boldsymbol{x}^T\boldsymbol{w}_i^* = t\right] + \epsilon d^{O(1)}$$

If $\boldsymbol{z}$ is away from every $\boldsymbol{w}_i^*$ by at least $\epsilon_2$ then again $\mathbb{E}[PC_3(f, \boldsymbol{z}, t)] = \mathbb{E}[P\delta'(\boldsymbol{x}^T\boldsymbol{w}_i^* - s)](s \in [t - \epsilon, t + \epsilon]) = \epsilon d^{O(1)}/\epsilon_2$. □

# D  STRUCTURAL RESTRICTIONS HELPS LEARNING

## D.1  PROOF OF THEOREM 7

To construct this correlation graph, we will run the following Algorithm 3 Denote $T_i := \{j : \boldsymbol{w}_{ij} = 1\}$. Let us compute $\mathbb{E}[f(x)x_i x_j]$:

$$\mathbb{E}[f(x)x_i x_j] = \sum_{S\subseteq[d]} c_S \mathbb{E}\left[\left(\prod_{p\in S} u(\boldsymbol{x}^T\boldsymbol{w}_p^*)\right)x_i x_j\right]$$
$$= \sum_{S\subseteq[d]} c_S e^{-\rho t|S|}\mathbb{E}\left[e^{\rho\sum_{p\in S}\boldsymbol{x}^T\boldsymbol{w}_p^* x_i x_j}\right]$$

---

**Algorithm 3** ConstructCorrelationGraph

---

1: Let $G$ be an undirected graph on $n$ vertices each corresponding to the $x_i$'s.
2: **for** every pair $i, j$ **do**
3:     Compute $\alpha_{ij} = \mathbb{E}[f(x)x_i x_j]$.
4:     **if** $\alpha_{ij} \geq \rho$ **then**
5:         Add edge $(i, j)$ to the graph.

---

$$
\begin{aligned}
&= \sum_{S \subseteq [d]} c_S e^{-\rho t |S|} \mathbb{E}\left[ e^{\rho \sum_{q \in \cup_{p \in S} T_p} x_q} x_i x_j \right] \\
&= \sum_{S \subseteq [d]} c_S e^{-\rho t |S|} \mathbb{1}[i, j \in \cup_{p \in S} T_p] \mathbb{E}\left[ x_i e^{\rho x_i} \right] \mathbb{E}\left[ x_j e^{\rho x_j} \right] \prod_{q \in \cup_{p \in S} T_p \backslash \{i,j\}} \mathbb{E}\left[ e^{\rho x_q} \right] \\
&= \sum_{S \subseteq [d]} c_S e^{-\rho t |S|} \mathbb{1}[i, j \in \cup_{p \in S} T_p] \rho^2 e^{\rho^2} \mathbb{E}\left[ x_j e^{\rho x_j} \right] \prod_{q \in \cup_{p \in S} T_p \backslash \{i,j\}} e^{\rho^2/2} \\
&= \sum_{S \subseteq [d]} c_S e^{-\rho t |S|} \mathbb{1}[i, j \in \cup_{p \in S} T_p] \rho^2 e^{\frac{\rho^2 |\cup_{p \in S} T_p|}{2}}
\end{aligned}
$$

By assumption, for all $p$, $T_p$ are disjoint. Now, if $i, j \in T_r$ for some $r$, we have

$$
\mathbb{E}[f(x)x_i x_j] = \rho^2 \sum_{S \subseteq [d] : r \in S} c_S e^{\frac{-\rho |S|(t - \rho d)}{2}}
$$

Similarly, if $i \in T_{r_1}$ and $j \in T_{r_2}$ with $r_1 \neq r_2$, we have

$$
\mathbb{E}[f(x)x_i x_j] = \rho^2 \sum_{S \subseteq [d] : r_1, r_2 \in S} c_S e^{\frac{-\rho |S|(t - \rho d)}{2}}
$$

It is easy to see that these correspond to coefficients of $X_r$ and $X_{r_1} X_{r_2}$ (respectively) in the following polynomial:

$$
Q(X_1, \ldots, X_n) = \rho^2 P(\mu(X_1 + 1), \ldots, \mu(X_n + 1))
$$

for $\mu = e^{\frac{-\rho(t - \rho d)}{2}}$. For completeness we show that this is true. We have,

$$
\begin{aligned}
Q(X_1, \ldots, X_n) &= \rho^2 \sum_{S \subseteq [d]} c_S \prod_{j \in S} \mu(X_j + 1) \\
&= \rho^2 \sum_{S \subseteq [d]} c_S \mu^{|S|} \prod_{j \in S} (X_j + 1)
\end{aligned}
$$

The coefficient of $X_r$ in the above form is clearly $\rho^2 \sum_{S \subseteq [d] : r \in S} c_S \mu^{|S|}$ (corresponds to picking the 1 in each product term). Similarly coefficient of $X_{r_1} X_{r_2}$ is $\rho^2 \sum_{S \subseteq [d] : r_1, r_2 \in S} c_S \mu^{|S|}$.

Now as in the assumptions, if we have a gap between these coefficients, then we can separate the large values from the small ones and form the graph of cliques. Each clique will correspond to the corresponding weight vector.

### D.2   PROOF OF THEOREM 8

Consider $f(\boldsymbol{x}) = \sum c_S \prod_{i \in S} e^{\boldsymbol{x}^T \boldsymbol{w}_i^*}$ where $\boldsymbol{w}_i^* = \sum_{j \in S_i} \boldsymbol{e}_j$ for $S_i \subseteq [n]$ such that for all $i \neq j$, $S_i \cap S_j = n$ and $c_S \geq 0$. Let us compute $g(\boldsymbol{z}) = e^{-\frac{||\boldsymbol{z}||^2}{2}} \mathbb{E}\left[ f(\boldsymbol{x}) e^{\boldsymbol{z}^T \boldsymbol{x}} \right]$ where $\boldsymbol{z} = \sum z_i \boldsymbol{e}_i$ for some $\alpha_i$.

$$
\mathbb{E}\left[ f(\boldsymbol{x}) e^{\boldsymbol{z}^T \boldsymbol{x}} \right] = \mathbb{E}\left[ \left( \sum c_S \prod_{i \in S} e^{\sum_{j \in S_i} \boldsymbol{x}_j} \right) e^{\boldsymbol{z}^T \boldsymbol{x}} \right]
$$

$$= \sum c_S \mathbb{E}\left[ \left( \prod_{i \in S} e^{\sum_{p \in S_i} \boldsymbol{x}_p} \right) e^{\sum_{q \in [n]} z_q \boldsymbol{x}_q} \right]$$

$$= \sum c_S \mathbb{E}\left[ \left( \prod_{p \in \cup_{i \in S} S_i} e^{(1+z_p)\boldsymbol{x}_p} \right) \left( \prod_{q \in [n] \setminus \cup_{i \in S} S_i} e^{z_q \boldsymbol{x}_q} \right) \right]$$

$$= \sum c_S \left( \prod_{p \in \cup_{i \in S} S_i} \mathbb{E}\left[ e^{(1+z_p)\boldsymbol{x}_p} \right] \right) \left( \prod_{q \in [n] \setminus \cup_{i \in S} S_i} \mathbb{E}\left[ e^{z_q \boldsymbol{x}_q} \right] \right)$$

$$= \sum c_S \left( \prod_{p \in \cup_{i \in S} S_i} e^{\frac{(1+z_p)^2}{2}} \right) \left( \prod_{q \in [n] \setminus \cup_{i \in S} S_i} e^{\frac{z_q^2}{2}} \right)$$

$$= \sum c_S e^{\frac{||\boldsymbol{z}||^2}{2}} \prod_{p \in \cup_{i \in S} S_i} e^{\frac{1}{2}+z_p}$$

$$\implies g(\boldsymbol{z}) = \sum c_S \prod_{p \in \cup_{i \in S} S_i} e^{\frac{1}{2}+z_p}$$

Consider the following optimization problem:

$$\max_{\boldsymbol{z}} \underbrace{\left[ g(\boldsymbol{z}) - \lambda ||\boldsymbol{z}||_1 - \gamma ||\boldsymbol{z}||_2^2 \right]}_{h(\boldsymbol{z})}$$

for $\lambda, \gamma > 0$ to be fixed later.

We can assume that $z_i \geq 0$ for all $i$ at a local maxima else we can move in the direction of $\boldsymbol{e}_i$ and this will not decrease $g(\boldsymbol{z})$ since $c_S \geq 0$ for all $S$ and will decrease $||\boldsymbol{z}||_1$ and $||\boldsymbol{z}||_2$ making $h(\boldsymbol{z})$ larger. From now on, we assume this for any $\boldsymbol{z}$ at local maxima.

We will show that the local maximas of the above problem will have $\boldsymbol{z}$ such that most of the mass is equally divided among $j \in S_i$ for some $i$ and close to 0 everywhere else.

**Lemma 21.** *There exists at most one $i$ such that there exists $j \in S_i$ with $|z_j| \geq \beta$ for $\gamma$ satisfying $4\gamma < \min_{i \neq j} \left[ e^\beta \sum_{S:i \in S, j \notin S \vee i \notin S, j \in S} c_S e^{\frac{|\cup_{i \in S} S_i|}{2}} \right]$.*

*Proof.* Let us prove by contradiction. Assume that there is a local maxima such that there are at least 2 indices say $1, 2$ such that $\exists j \in S_1, k \in S_2, |z_j|, |z_k| \geq \beta$. Now we will show that there exists a perturbation such that $g(\boldsymbol{z})$ can be improved. Now consider the following perturbation, $\boldsymbol{z} + s\epsilon \boldsymbol{e}_j - s\epsilon \boldsymbol{e}_k$ for $s \in \{\pm 1\}$. Observe that $||\boldsymbol{z}||_1$ remains unchanged for $\epsilon < \beta$ also $||\boldsymbol{z}||_2^2$ changes by $2s^2\epsilon^2 + 2(z_j - z_k)s\epsilon$. We have

$$\mathbb{E}_s[h(\boldsymbol{z} + s\epsilon \boldsymbol{e}_j - s\epsilon \boldsymbol{e}_k) - h(\boldsymbol{z})]$$

$$= \sum_{S:1 \in S, 2 \notin S} c_S \prod_{p \in \cup_{i \in S} S_i} e^{\frac{1}{2}+z_p} \left( \mathbb{E}_s\left[ e^{s\epsilon} \right] - 1 \right) + \sum_{S:1 \notin S, 2 \in S} c_S \prod_{p \in \cup_{i \in S} S_i} e^{\frac{1}{2}+z_p} \left( \mathbb{E}_s\left[ e^{-s\epsilon} \right] - 1 \right)$$

$$- \gamma \mathbb{E}_s\left[ 2\epsilon^2 + 2(z_j - z_k)s\epsilon \right]$$

$$\geq \frac{\epsilon^2}{2} \left( -4\gamma + \sum_{S:1 \in S, 2 \notin S} c_S \prod_{p \in \cup_{i \in S} S_i} e^{\frac{1}{2}+z_p} + \sum_{S:1 \notin S, 2 \in S} c_S \prod_{p \in \cup_{i \in S} S_i} e^{\frac{1}{2}+z_p} \right)$$

The inequality follows since $\mathbb{E}\left[ e^{s\epsilon} \right] = \frac{e^\epsilon + e^{-\epsilon}}{2} \geq 1 + \frac{\epsilon^2}{2}$. Observe that

$$\sum_{S:1 \in S, 2 \notin S} c_S \prod_{p \in \cup_{i \in S} S_i} e^{\frac{1}{2}+z_p} \geq \sum_{S:1 \in S, 2 \notin S} c_S e^{\frac{|\cup_{i \in S} S_i|}{2}} e^{z_j} \geq \sum_{S:1 \in S, 2 \notin S} c_S e^{\frac{|\cup_{i \in S} S_i|}{2}} e^\beta.$$

For chosen value of $\gamma$, there will always be an improvement, hence can not be a local maxima. $\square$

**Lemma 22.** *At the local maxima, for all $i \in [n]$, $\boldsymbol{z}$ is such that for all $j, k \in S_i$, $z_j = z_k$ at local maxima.*

*Proof.* We prove by contradiction. Suppose there exists $j, k$ such that $z_j < z_k$. Consider the following perturbation: $\boldsymbol{z} + \epsilon(z_k - z_j)(\boldsymbol{e}_j - e_k)$ for $1 \leq \epsilon > 0$. Observe that $g(\boldsymbol{z})$ depends on only $\sum_{r \in S_i} z_r$ and since that remains constant by this update $g(\boldsymbol{z})$ does not change. Also note that $||\boldsymbol{z}||_1$ does not change. However $||\boldsymbol{z}||_2^2$ decreases by $2\epsilon(1-\epsilon)(z_k - z_j)^2$ implying that overall $h(\boldsymbol{z})$ increases. Thus there is a direction of improvement and thus it can not be a local maxima. $\square$

**Lemma 23.** *At the local maxima, $||\boldsymbol{z}||_1 \geq \alpha$ for $\lambda < \sum_S c_S \frac{|\cup_{i \in S} S_i|}{n} e^{\frac{|\cup_{i \in S} S_i|}{2}} - \gamma(2\alpha + 1)$.*

*Proof.* We prove by contradiction. Suppose $||\boldsymbol{z}||_1 < \alpha$, consider the following perturbation, $\boldsymbol{z} + \epsilon \mathbb{1}$. Then we have

$$
h(\boldsymbol{z} + \epsilon \mathbb{1}) - h(\boldsymbol{z}) = \sum c_S e^{\frac{|\cup_{i \in S} S_i|}{2} + \sum_{p \in \cup_{i \in S} S_i} z_p} (e^{\epsilon|\cup_{i \in S} S_i|} - 1) - n\lambda\epsilon - n\gamma\epsilon(2||\boldsymbol{z}||_1 + \epsilon)
$$

$$
> \sum c_S e^{\frac{|\cup_{i \in S} S_i|}{2}} |\cup_{i \in S} S_i|\epsilon - n\lambda\epsilon - n\gamma\epsilon(2\alpha + 1)
$$

$\square$

For given $\lambda$ there is a direction of improvement giving a contradiction that this is the local maxima.

Combining the above, we have that we can choose $\lambda, \gamma = \mathsf{poly}(n, 1/\epsilon, s)$ where $s$ is a paramater that depends on structure of $f$ such that at any local maxima there exists $i$ such that for all $j \in S_i$, $z_j \geq 1$ and for all $k \notin \cup_{j \in S_i}, z_k \leq \epsilon$.

### D.3 PROOF OF THEOREM 9

Let function $f = \sum_{S \subseteq [n]} c_S \prod_{i \in S} u((\boldsymbol{w}_i^*)^T x)$ for orthonormal $\boldsymbol{w}_i^*$. WLOG, assume $\boldsymbol{w}_i^* = \boldsymbol{e}_i$. Let $u(x) = \sum_{i=0}^{\infty} \hat{u}_{2i} h_{2i}(x)$ where $h_i$ are hermite polynomials and $u'(x) = u(x) - \hat{u}_0 = \sum_{i=1}^{\infty} \hat{u}_{2i} h_{2i}(x)$. This implies $\mathbb{E}[u'(x)] = 0$. Observe that,

$$
\prod_{i \in S} u(x_i) = \prod_{i \in S} (u'(x_i) + \hat{u}_0) = \sum_{k=0}^{|S|} \hat{u}_0^{|S|-k} \sum_{S' \subseteq S: |S'|=k} \prod_{i \in S'} u'(x_i).
$$

Let us consider correlation with $h_4(\boldsymbol{z}^T \boldsymbol{x})$. This above can be further simplified by observing that when we correlate with $h_4$, $\prod_{i \in S'} u'(x_i)h_4(\boldsymbol{z}^T \boldsymbol{x}) = 0$ for $|S'| \geq 2$. Observe that $h_4(\boldsymbol{z}^T \boldsymbol{x}) = \sum_{d_1,\ldots,d_n \in [4]: \sum d_i \leq 4} c(d_1, \ldots, d_n) \prod h_{d_i}(x_i)$ for some coefficients $c$ which are functions of $\boldsymbol{z}$. Thus when we correlate $\prod_{i \in S'} u'(x_i) h_4(\boldsymbol{z}^T \boldsymbol{x})$ for $|S'| \geq 3$ then we can only get a non-zero term if we have at least $h_{2k}(x_i)$ with $k \geq 1$ for all $i \in S'$. This is not possible for $|S'| \geq 3$, hence, these terms are 0. Thus,

$$
\mathbb{E}\left[\prod_{i \in S} u(x_i) h_4(\boldsymbol{z}^T \boldsymbol{x})\right] = \sum_{k=0}^{2} \hat{u}_0^{|S|-k} \sum_{S' \subseteq S: |S'|=k} \mathbb{E}\left[\prod_{i \in S'} u'(x_i) h_4(\boldsymbol{z}^T \boldsymbol{x})\right].
$$

Lets compute these correlations.

$$
\mathbb{E}\left[u'(x_i) h_4(\boldsymbol{z}^T \boldsymbol{x})\right]
$$

$$
= \mathbb{E}\left[\sum_{p>0} \hat{u}_{2p} h_{2p}(x_i) h_4(z_i x_i + z_{-i}^T \boldsymbol{x}_{-i})\right]
$$

$$
= \frac{1}{4} \sum_{p>0} \hat{u}_{2p} \mathbb{E}\left[h_{2p}(x_i) \sum_{k=0}^{4} \binom{4}{k} h_{4-k}(z_i x_i \sqrt{2}) h_k(z_{-ij}^T \boldsymbol{x}_{-ij} \sqrt{2})\right]
$$

$$
= \frac{1}{4} \sum_{p>0} \hat{u}_{2p} \mathbb{E}\left[h_{2p}(x_i) \left(h_4(z_i x_i \sqrt{2}) + 6h_2(z_i x_i \sqrt{2})(2||z_{-i}||^2 - 1) + 3(2||z_{-i}||^2 - 1)^2\right)\right]
$$

$$= \frac{1}{4}\left(\hat{u}_4\alpha(4,4,z_i\sqrt{2}) + \hat{u}_2\alpha(4,2,z_i\sqrt{2}) + 6\hat{u}_2\alpha(2,2,z_i\sqrt{2})(2||z_{-i}||^2 - 1)\right)$$

$$= \frac{1}{4}\left(4\hat{u}_4 z_i^4 + 12\hat{u}_2 z_i^2(2z_i^2 - 1) + 6\hat{u}_2(2z_i^2)(2||z_{-i}||^2 - 1)\right)$$

$$= \hat{u}_4 z_i^4 + 3\hat{u}_2 z_i^2(2||\boldsymbol{z}||^2 - 1)$$

$\mathbb{E}\left[u'(x_i)u'(x_j)h_4(\boldsymbol{z}^T\boldsymbol{x})\right]$

$$= \mathbb{E}\left[\sum_{p,q>0}\hat{u}_{2p}\hat{u}_{2q}h_{2p}(x_i)h_{2q}(x_j)h_4(z_i x_i + z_j x_j + z_{-ij}^T\boldsymbol{x}_{-ij})\right]$$

$$= \frac{1}{4}\sum_{p,q>0}\hat{u}_{2p}\hat{u}_{2q}\mathbb{E}\left[h_{2p}(x_i)h_{2q}(x_j)\sum_{k=0}^{4}\binom{4}{k}h_{4-k}((z_i x_i + z_j x_j)\sqrt{2})h_k(z_{-ij}^T\boldsymbol{x}_{-ij}\sqrt{2})\right]$$

$$= \frac{1}{4}\sum_{p,q>0}\hat{u}_{2p}\hat{u}_{2q}\mathbb{E}\left[h_{2p}(x_i)h_{2q}(x_j)\left(h_4((z_i x_i + z_j x_j)\sqrt{2}) + 6h_2((z_i x_i + z_j x_j)\sqrt{2})(2||z_{-ij}||^2 - 1)\right)\right.$$

$$\left. + 3(2||z_{-ij}||^2 - 1)^2\right)\Big]$$

$$= \frac{1}{16}\underbrace{\sum_{p,q}\hat{u}_p\hat{u}_q\sum_{k=0}^{4}\binom{4}{k}\mathbb{E}\left[h_{2p}(x_i)h_{2q}(x_j)h_{4-k}(2z_i x_i)h_k(2z_j x_j)\right]}_{\textstyle{\textcircled{\scriptsize 1}}}$$

$$+ \underbrace{\frac{3}{4}(2||z_{-ij}||^2 - 1)\sum_{p,q<0}\hat{u}_{2p}\hat{u}_{2q}\sum_{k=0}^{2}\binom{2}{k}\mathbb{E}\left[h_{2p}(x_i)h_{2q}(x_j)h_{2-k}(2z_i x_i)h_k(2z_j x_j)\right]}_{\textstyle{\textcircled{\scriptsize 2}}}$$

We will compute $\textcircled{\scriptsize 1}$ and $\textcircled{\scriptsize 2}$:

$$\textcircled{\scriptsize 1} = \frac{1}{16}\sum_{k=0}^{4}\binom{4}{k}\sum_{p,q>0}\hat{u}_{2p}\hat{u}_{2q}\mathbb{E}\left[h_{2p}(x_i)h_{4-k}(2z_i x_i)\right]\mathbb{E}\left[h_{2q}(x_j)h_k(2z_j x_j)\right]$$

$$= \frac{1}{16}\sum_{k=0}^{4}\binom{4}{k}\sum_{p,q>0}\hat{u}_{2p}\hat{u}_{2q}\alpha(4-k,2p,2z_i)\alpha(k,2q,2z_j)$$

$$= \frac{3}{8}\hat{u}_2^2\alpha(2,2,2z_i)\alpha(2,2,2z_j)$$

$$= 6\hat{u}_2^2 z_i^2 z_j^2$$

Similarly,

$$\textcircled{\scriptsize 2} = \frac{3}{4}(2||z_{-ij}||^2 - 1)\sum_{p,q>0}\hat{u}_{2p}\hat{u}_{2q}\sum_{k=0}^{2}\binom{2}{k}\mathbb{E}\left[h_{2p}(x_i)h_{2q}(x_j)h_{2-k}(2z_i x_i)h_k(2z_j x_j)\right]$$

$$= \frac{3}{4}(2||z_{-ij}||^2 - 1)\sum_{k=0}^{2}\binom{2}{k}\hat{u}_2\hat{u}_2\alpha(2-k,2,2z_i)\alpha(k,2,2z_j) = 0.$$

Combining, we get

$$\mathbb{E}\left[u'(x_i)u'(x_j)h_4(\boldsymbol{z}^T\boldsymbol{x})\right] = 6\hat{u}_2^2 z_i^2 z_j^2.$$

Further, taking correlation with $f$, we get:

$\mathbb{E}\left[f(x)h_4(\boldsymbol{z}^T\boldsymbol{x})\right]$

$$= \sum_{S\subseteq[n]}c_S\sum_{k=0}^{2}\hat{u}_0^{|S|-k}\sum_{S'\subseteq S:|S'|=k}\mathbb{E}\left[\prod_{i\in S'}u'(x_i)h_4(\boldsymbol{z}^T\boldsymbol{x})\right]$$

$$= \sum_{S \subseteq [n]} c_S \hat{u}_0^{|S|-2} \left( \hat{u}_0 \sum_{i \in S} (\hat{u}_4 z_i^4 + 3\hat{u}_2 z_i^2 (2||\boldsymbol{z}||^2 - 1)) + 6\hat{u}_2^2 \sum_{j \neq k \in S} z_j^2 z_k^2 \right) + \text{constant}$$

$$= \sum_{i=1}^{n} \alpha_i z_i^4 + (2||\boldsymbol{z}||^2 - 1) \sum_{i=1}^{n} \beta_i z_i^2 + \sum_{1 \leq i \neq j \leq n} \gamma_{ij} z_j^2 z_k^2 + \text{constant}$$

$$= \sum_{i=1}^{n} \alpha_i z_i^4 + \sum_{i=1}^{n} \beta_i z_i^2 + \sum_{1 \leq i \neq j \leq n} \gamma_{ij} z_j^2 z_k^2 + \text{constant}$$

where $\alpha_i = \hat{u}_4 \sum_{S' \subseteq [n] | i \in S'} c_{S'} \hat{u}_0^{|S'|-1}$, $\beta_i = 3\hat{u}_2 \sum_{S' \subseteq [n] | i \in S'} c_{S'} \hat{u}_0^{|S'|-1}$ and $\gamma_{ij} = 6\hat{u}_2^2 \sum_{S' \subseteq [n] | i,j \in S'} c_{S'} \hat{u}_0^{|S'|-2}$.

If $\gamma_{ij} < \alpha_i + \alpha_j$ for all $i, j$ then the local maximas of the above are exactly $\boldsymbol{e}_i$. To show that this holds, we prove by contradiction. Suppose there is a maxima where $z_i, z_j \neq 0$. Then consider the following second order change $z_i^2 \to z_i^2 + s\epsilon$ and $z_j^2 \to z_j^2 - s\epsilon$ where $\epsilon \leq \min z_i^2, z_j^2$ and $s$ is 1 with probability 0.5 and -1 otherwise. Observe that the following change does not violate the constraint and in expectation affects the objective as follows:

$$\Delta = \mathbb{E}_s \left[ \alpha_i(2s\epsilon z_i^2 + \epsilon^2) + \alpha_j(-2s\epsilon z_i^2 + \epsilon^2) + \beta_i s\epsilon - \beta_j s\epsilon + \gamma_{ij}(s\epsilon z_j^2 - s\epsilon z_i^2 - \epsilon^2) \right]$$
$$= (\alpha_i + \alpha_j - \gamma_{ij})\epsilon^2 > 0$$

Thus there is a direction in which we can improve and hence it can not be a maxima.

# E  MODULAR LEARNING BY DIVIDE AND CONQUER

Finally we point out how such high threshold layers could potentially facilitate the learning of deep functions $f$ at any depth, not just at the lowest layer. Note that essentially for Lemma 2 to hold, outputs $X_1, ., X_d$ needn't be present after first layer but they could be at any layer. If there is any cut in the network that outputs $X_1, ... X_d$, and if the upper layer functions can be modelled by a polynomial $P$, then again assuming the inputs $X_i$ have some degree of independence one can get something similar to Lemma 2 that bounds the non-linear part of $P$. The main property we need is $E[\Pi_{i \in S} X_i]/E[X_j] < \mu^{|S|-1}$ for a small enough $\mu = 1/\text{poly}(d)$ which is essentially replaces Lemma 1. Thus high threshold layers can essentially reduce the complexity of learning deep networks by making them roughly similar to a network with lower depth. Thus such a cut essentially divides the network into two simpler parts that can be learned separately making it amenable to a divide and conquer approach. If there a robust algorithm to learn the lower part of the network that output $X_i$, then by training the function $f$ on that algorithm would recover the lower part of the network, having learned which one would be left with learning the remaining part $P$ separately.

**Remark 1.** *If there is a layer of high threshold nodes at an intermediate depth $l$, $u$ is sign function, if outputs $X_i$ at depth $l$ satisfy the following type of independence property: $E[\Pi_{i \in S} X_i]/E[X_j] < \mu^{|S|-1}$ for a small enough $\mu = 1/\text{poly}(d)$, if there is a robust algorithm to learn $X_i$ from $\sum c_i X_i$ that can tolerate noise, then one can learn the nodes $X_i$, from a function $P(X_1, .., X_d)$*

