# OpenReview forum: "Recovering the Lowest Layer of Deep Networks with High Threshold Activations"
_ICLR.cc/2019/Conference_

### Official Review · AnonReviewer3 · 2018-11-02

**Rating:** 4
**Confidence:** 4

**Review:**

This paper gives a new algorithm for learning parameters of neural network under several assumptions: 1. the threshold for the first layer is very high; 2. the future layers of the neural network can be approximated by a polynomial. 3. The input distribution is Gaussian.

It is unclear why any of these assumptions are true. For 1, the thresholds in neural networks are certainly not as high as required in the algorithm (for the threshold in the paper after the first layer the neurons will be super sparse/often even just equal to 0, this is not really observed in real neural networks). For 2, there are no general results showing neural networks can be effectively approximated by low degree polynomials, and, if the future networks can be approximated, what prevents you from just assuming the entire neural network is a low degree polynomial? People have tried fitting polynomials and that does not perform nearly as well as neural networks.

The proof of the paper makes the problem even more clear because the paper shows that with this high threshold in the first layer, the future layers just behave linearly. This is again very far from true in any real neural networks.

Overall I'm OK with making some strong assumptions in order to prove some results for neural networks - after all it is a very difficult problem. However, this paper makes too many unrealistic assumptions. It's OK to make one of these assumptions, maybe 2, but 3 is too much for me.

---

> ### Author Response · Authors · 2018-11-20
> **response**
>
> We would like to thank the reviewer for their comments. We would like to clarify the assumptions:
> 1) We do not make a low degree assumption. For ease of presentation we assume that the polynomial has monomials with each variable having degree 1. Overall the degree can still be d. In the Appendix B.8, we extend the same to higher degree with a stronger assumption on the coefficients of the polynomial.
>
> 2) As for high thresholds at the first layer, we agree that high thresholds may not be present at the first layer -- but they tend to be present in the higher layers. Even though we assume this at the first layer we point out in section E how this idea may apply at a higher layer. Thus our overall point is that, if we have an algorithm for learning a k-hidden layer network and there are high thresholds at the kth layer, then we may still be able to recover it approximately with more layers over it.
>
> 3) We note that the function given by the upper layers is close to linear layers only in expectation, not necessarily point-wise. This closeness may not be very small if d is viewed as a constant. Also note that even though it is close to linear, we are able to learn the first layer to any precision for monotone P.
> Also note that even though the function may look like a simple two layer network with linearity on top most of the time, on a small fraction of inputs it is doing something more sophisticated than a 2 layer network -- its full power is visible only an a small fraction of the inputs in the distribution and it is this full power that is manifested less often that we are interested in learning. Our goal is in fact to learn the entire network layer by layer. What we are showing here is that given a complex multi-layer network with high threshold, it automatically makes it look close to a 2 layer network (most of the time) and so we can learn the first layer. If one could proceed this way to second, third layer and so on, we would in effect learn the entire complex network. Thus we are saying that a high threshold network may make it easier to learn the network layer by layer. Now it is true that so far we have only managed to learn the first layer -- but we believe this method may be extendable to other layers too.
>
> We hope the reviewer will take the above discussion into consideration.

---

### Official Review · AnonReviewer5 · 2018-11-12

**Rating:** 5
**Confidence:** 3

**Review:**

This paper considers the problem of recovering the lowest layer of a deep neural network whose architecture is ReLU or sign function followed by a polynomial. This paper relies on three assumptions: 1) the lowest layer has a high threshold (\Omleg(\sqrt{d})), 2) the polynomial has 1/poly(d) lower bouned and O(1) upper bounded linear terms and is monotone 3) the input is Gaussian. Under these assumptions, this paper shows it is possible to learn the lowest layer in precision \eps in poly(1/eps, d) time.

The proposed algorithm has two steps. The first step is based on the landscape design approach proposed by Ge et al. (2017) and the second step is based on checking the correlation.

Provably learning a neural network is a major problem in theoretical machine learning. The assumptions made in this paper are fine for me and I think this paper indeed has some new interesting observation. My major concern is the writing. There are several components of the algorithm. However, it is hard to digest the intuition behind each component and how the assumptions are used. I suggest authors providing a high-level and non-technical description of the whole algorithm at the beginning. If authors can significantly improve the writing, I am happy to re-evaluate my comments and increase my rating.

---

> ### Author Response · Authors · 2018-11-20
> **response**
>
> We would like to thank the reviewer for the constructive feedback. Following the reviewer's advice, we have added a high-level overview of the algorithm before describing the technical content (beginning of section 2).
>
> We hope the reviewer will take this into consideration.

---

### Official Review · AnonReviewer4 · 2018-11-14

**Rating:** 4
**Confidence:** 4

**Review:**

This paper gives provable recovery guarantees for a class of neural networks which have high-threshold activation in the first layer, followed by a "well-behaved" polynomial, under Gaussian input. The algorithm is based on the approach by Ge et al. (2017), as well as an iterative refinement method.

While this could be an interesting result, I have several concerns regarding the assumptions, correctness, and writing.

1) It is required that the threshold is at least sqrt{log d} (Thm. 1), where d is the number of hidden neurons in the first layer. It seems that this essentially zeros out almost all the neurons, since the maximum among d Gaussian random variables is roughly sqrt{log d}. The authors should explain what exactly this model is doing, i.e., what kind of functions it can compute, in order to justify why this is an interesting model.

Furthermore, the authors claim that the studied model is a "deep" neural network, but I disagree. As I understand, the difference between this model and two-layer networks is that the second layer here is a polynomial instead of a linear function. This doesn't make it a deep network since the (polynomial) part above the first layer is not modeled in a layer-wise fashion, not to mention that under the setting considered in the paper the polynomial behaves similar to a linear function.

2) It is stated at the end of Section 2 that the angle can be reduced by a factor of 1-1/d ***with constant probability***. How does this ensure you can succeed after O(d log(1/nu)) iterations? As far as I see you need the success probability in one iteration to be at least something like 1-1/Omega(d) so that you can apply a union bound.

3) Even if the issues of motivation and correctness are clarified, I find it very difficult to understand the overall intuition and main technical contributions in this paper. The writing needs to be significantly improved to reach the level of a top conference.

---

> ### Author Response · Authors · 2018-11-20
> **response**
>
> We would like to thank the reviewer for their feedback. We would like to answer for each point:
> 1) a) We agree that the threshold seems high however we note that this might be necessary for the following reason. Our work can be viewed in the SIGN activation case as learning a union of half spaces. Conditioning on the label being positive, the distribution of the points can be viewed as being drawn from a mixture of halfspaces. The distribution of points on the positive side of one hyperplane can be viewed as a truncated Gaussian distribution with mean as the normal to the halfspace. With this viewpoint, recovering the weight vectors corresponding to the union of halfspaces is intuitively similar to learning the means of a mixture of Gaussians.  The threshold value of Omega(\sqrt \log d) matches the bound between centers required to learn a mixture of Gaussians. The paper https://arxiv.org/abs/1710.11592 (FOCS 2017) shows a lower bound of Omega(\sqrt \log d) to be able to recover a collection of d Gaussians in polynomial sample complexity. In fact, for a mixture of Gaussians the FOCS 2017 paper only gives a polynomial sample complexity upper bound with exponential running time whereas for our problem we recover the half spaces in polynomial time. Also note that, as a by product, we are getting the first polynomial algorithm to learn intersection of halfspaces if they are \Omega(\sqrt \log d) far from the origin.
> b) As for the criticism that the polynomial is close to a linear function, please see bullet 3 in response to AnonReviewer3.
> c) As for why our framework of using a polynomial P applies to a deep network: Note that if the activation function is continuous such as sigmoid or log(1+e^x) (which is a continuous approximation of RELU) it will have a Taylor series, and further, stacking such activations layer by layer will also have a Taylor series. All we are assuming is that there is a Taylor series that models the function represented by the upper layers. We are just calling that Taylor representation as P.
>
> 2) With a constant probability we can improve after each iteration, however we can also check if we have improved or not by using our estimating technique, so we only change our estimate if we improve. We refer the reviewer to step 3 & 4 of Algorithm 1. Thus we have to try only a constant times to get a good move and make progress, hence the bound.
>
> We hope the reviewer will take the above discussion into consideration.

---

### Meta-Review · Area_Chair1 · 2018-12-19

**Recommendation:** Reject
**Confidence:** 5

**Metareview:**

The reviewers reached a consense on that the paper is not quite ready for publication at ICRL. The main potential drawback include a) the exposition of the paper can be improved; b) it's not entirely clear that some of the assumptions (such as the threshold for the first layer, the polynomial approximation of higher layers) are meaningful , and it seems that the proof technique exploits heavily some of these assumptions and some of the key intermediate steps won't hold in practice. (see reviewer 3's comment for more details.) The authors clarify the writing and intuitions in the response, but overall the AC decided that the paper is not quite ready for publications at the moment.